# PRIOR MISMATCH AND ADAPTATION IN PNP-ADMM WITH A NONCONVEX CONVERGENCE ANALYSIS

## ABSTRACT

Plug-and-Play (PnP) priors is a widely-used family of methods for solving imaging inverse problems by integrating physical measurement models with image priors specified using image denoisers. PnP methods have been shown to achieve state-of-the-art performance when the prior is obtained using powerful deep denoisers. Despite extensive work on PnP, the topic of *distribution mismatch* between the training and testing data has often been overlooked in the PnP literature. This paper presents a set of new theoretical and numerical results on the topic of prior distribution mismatch and domain adaptation for the *alternating direction method of multipliers (ADMM)* variant of PnP. Our theoretical result provides an explicit error bound for PnP-ADMM due to the mismatch between the desired denoiser and the one used for inference. Our analysis contributes to the work in the area by considering the mismatch under *nonconvex* data-fidelity terms and *expansive* denoisers. Our first set of numerical results quantifies the impact of the prior distribution mismatch on the performance of PnP-ADMM on the problem of image super-resolution. Our second set of numerical results considers a simple and effective domain adaption strategy that closes the performance gap due to the use of mismatched denoisers. Our results suggest the relative robustness of PnP-ADMM to prior distribution mismatch, while also showing that the performance gap can be significantly reduced with few training samples from the desired distribution.

## 1 INTRODUCTION

*Imaging inverse problems* consider the recovery of a clean image from its corrupted observation. Such problems arise across the fields of computational imaging, biomedical imaging, and computer vision. As imaging inverse problems are typically ill-posed, solving them requires the use of image priors. While many approaches have been proposed for implementing image priors, the current literature is primarily focused on methods based on training *deep learning (DL)* models to map noisy observations to clean images (McCann et al., 2017; Lucas et al., 2018; Ongie et al., 2020).

*Plug-and-Play (PnP)* Priors (Venkatakrishnan et al., 2013; Sreehari et al., 2016) have emerged as a class of iterative algorithms that can use DL denoisers as implicit image priors for solving inverse problems. PnP algorithms sequentially minimize a data-fidelity term to improve data consistency and then perform regularization through an image denoiser. PnP has been successfully used in many applications such as super-resolution, phase retrieval, microscopy, and medical imaging (Metzler et al., 2018; Zhang et al., 2017; Meinhardt et al., 2017; Dong et al., 2019; Zhang et al., 2019; Wei et al., 2020; Zhang et al., 2021). The success of PnP has resulted in the development of its multiple variants (e.g., PnP-PGM, PnP-SGD, PnP-ADMM. PnP-HQS) (Romano et al., 2017; Buzzard et al., 2018; Yuan et al., 2020; Reehorst & Schniter, 2019; Hurault et al., 2022a; Kamilov et al., 2023), strong interest in its theoretical analysis (Chan et al., 2017; Teodoro et al., 2019; Ahmad et al., 2020; Sun et al., 2019b;a; Liu et al., 2021), as well as investigation of its connection to other methods used in inverse problems, such as score matching (Cohen et al., 2021; Reehorst & Schniter, 2019) and denoising diffusion probabilistic models (Kadkhodaie & Simoncelli, 2021; Laumont et al., 2022).

Despite extensive literature on PnP, the research in the area has mainly focused on the setting where the distribution of the inference data is perfectly matched to that of the data used for training deep learning denoisers, used as image priors in PnP. Little work exists for PnP under mismatched deep learning-based priors, where a distribution shift exists between the training and test data (Shoushtari

**Figure 1:** *Illustration of domain adaptation in PnP-ADMM. The mismatched denoiser is pre-trained on source distribution (BreCaHAD) and adapted to target distribution (MetFaces) using a few samples. Adapted prior is then plugged into PnP-ADMM algorithm to reconstruct a sample from MetFaces.*

et al., 2022; Reehorst & Schniter, 2019). In this paper, we investigate the problem of *mismatched* priors in PnP-ADMM. We present a new theoretical analysis of PnP-ADMM that accounts for the use of mismatched priors. Unlike most existing work on PnP-ADMM, our theory is compatible with *nonconvex* data-fidelity terms and *expansive* denoisers (Sun et al., 2021; Tang & Davies, 2020; Gavaskar & Chaudhury, 2019; Chan, 2019; Ryu et al., 2019). Our analysis establishes explicit error bounds on the convergence of PnP-ADMM under a well-defined set of assumptions. We validate our theoretical findings by presenting numerical results on the influence of distribution shifts, where the denoiser trained on one dataset (e.g., BreCaHAD or CelebA) is used to recover an image from another dataset (e.g., MetFaces or RxRx1). We additionally present numerical results on a simple domain adaptation strategy for image denoisers that can effectively address data distribution shifts in PnP methods (see Figure 4 for an illustration). Our work thus enriches the current PnP literature by providing novel theoretical and empirical insights into the problem of data distribution shifts in PnP.

All proofs and some details that have been omitted due to space constraints of the main text are included in the supplementary material.

## 2 BACKGROUND

**Inverse problems.** Inverse problems involve the recovery of an unknown signal $\boldsymbol{x} \in \mathbb{R}^n$ from a set of noisy measurements $\boldsymbol{y} = \boldsymbol{A}\boldsymbol{x} + \boldsymbol{e}$, where $\boldsymbol{A} \in \mathbb{R}^{m \times n}$ is the measurement model and $\boldsymbol{e}$ is the noise. Inverse problems are often formulated and solved as optimization problems of form

$$\widehat{\boldsymbol{x}} \in \underset{\boldsymbol{x} \in \mathbb{R}^n}{\arg\min}\, f(\boldsymbol{x}) \quad \text{with} \quad f(\boldsymbol{x}) = g(\boldsymbol{x}) + h(\boldsymbol{x})\,, \tag{1}$$

where $g$ is the data-fidelity term that measures the consistency with the measurements $\boldsymbol{y}$ and $h$ is the regularizer that incorporates prior knowledge on $\boldsymbol{x}$. The least-squares function $g(\boldsymbol{x}) = \frac{1}{2}\|\boldsymbol{A}\boldsymbol{x} - \boldsymbol{y}\|_2^2$ and total variation (TV) function $h(\boldsymbol{x}) = \tau\|\boldsymbol{D}\boldsymbol{x}\|_1$, where $\boldsymbol{D}$ denotes the image gradient and $\tau > 0$ a regularization parameter, are commonly used functions for the data-fidelity term and the regularizer (Rudin et al., 1992; Beck & Teboulle, 2009).

**Deep Learning.** DL has gained significant attention in the context of inverse problems. DL methods seek to perform a regularized inversion by learning a mapping from the measurements to the target images parameterized by a deep convolutional neural network (CNN) (McCann et al., 2017; Lucas et al., 2018; Ongie et al., 2020). Model-based DL (MBDL) refers to a sub-class of DL methods for inverse problems that also integrate the measurement model as part of the deep model (Ongie et al., 2020; Monga et al., 2021). A class of MBDL that incorporates deep denoisers as implicit image priors within iterative algorithms includes PnP, regularization by denoising (RED), deep unfolding (DU), and deep equilibrium models (DEQ) (Zhang & Ghanem, 2018; Hauptmann et al., 2018; Gilton et al., 2021; Liu et al., 2022).

**Plug-and-Play Priors.** PnP is a popular MBDL approach for solving imaging inverse problems by using denoisers as image priors within iterative algorithms (Venkatakrishnan et al., 2013) (see also recent reviews (Ahmad et al., 2020; Kamilov et al., 2023)). Motivated by proximal splitting algorithms, PnP can replace the proximal or gradient descent updates with deep denoisers while simultaneously minimizing the data-fidelity function to recover consistent solutions. PnP methods can thus be viewed as MBDL architectures that integrate measurement models and deep denoisers. PnP has been extensively investigated, leading to multiple PnP variants and theoretical analyses (Chan

et al., 2017; Buzzard et al., 2018; Sun et al., 2021; Ryu et al., 2019; Hurault et al., 2022a; Laumont et al., 2022; Tirer & Giryes, 2019; Teodoro et al., 2019; Sun et al., 2019b; Cohen et al., 2021). Existing theoretical convergence analyses of PnP differ in the specifics of the assumptions required to ensure the convergence of the corresponding iterations. For example, bounded, averaged, firmly nonexpansive, nonexpansive, residual nonexpansive, or demi-contractive denoisers have been previously considered for designing convergent PnP schemes (Chan et al., 2017; Gavaskar & Chaudhury, 2019; Romano et al., 2017; Ryu et al., 2019; Cohen et al., 2021; Sun et al., 2019a; 2021; Terris et al., 2020; Reehorst & Schniter, 2019; Liu et al., 2021; Hertrich et al., 2021; Bohra et al., 2021). The recent work (Xu et al., 2020) has used an elegant formulation of an MMSE denoiser from (Gribonval, 2011) to perform a nonconvex convergence analysis of PnP-PGM without any nonexpansiveness assumptions on the denoiser. Another recent line of PnP work has explored specification of the denoiser as a gradient-descent step on a functional parameterized by a deep neural network (Hurault et al., 2022a;b; Cohen et al., 2021).

PnP-ADMM is summarized in Algorithm 1 (Sreehari et al., 2016; Venkatakrishnan et al., 2013), where $\mathsf{D}_\sigma$ is an additive white Gaussian denoiser (AWGN) denoiser, $\gamma > 0$ is the penalty parameter, and $\sigma > 0$ controls the denoiser strength. PnP-ADMM is based on the alternating direction method of multipliers (ADMM) (Boyd et al., 2011). Its formulation relies on optimizing in an alternating fashion the augmented Lagrangian associated with the objective function in (1)

$$\phi\left(\boldsymbol{x}, \boldsymbol{z}, \boldsymbol{s}\right) = g(\boldsymbol{x}) + h(\boldsymbol{z}) + \frac{1}{\gamma}\boldsymbol{s}^\mathsf{T}(\boldsymbol{x} - \boldsymbol{z}) + \frac{1}{2\gamma}\left\|\boldsymbol{x} - \boldsymbol{z}\right\|_2^2. \tag{2}$$

The theoretical convergence of PnP-ADMM has been explored for convex functions using monotone operator theory (Ryu et al., 2019; Sun et al., 2021), for nonconvex regularizer and convex data-fidelity terms (Hurault et al., 2022b), and for bounded denoisers (Chan et al., 2017).

**Distribution Shift.** Distribution shifts naturally arise in imaging when a DL model trained on one type of data is applied to another. The mismatched DL models due to distribution shifts lead to suboptimal performance. Consequently, there has been interest in mitigating the effect of mismatched DL models (Sun et al., 2020; Darestani et al., 2021; 2022; Jalal et al., 2021). In PnP methods, a mismatch arises when the denoiser is trained on a distribution different from that of the test data. The prior work on denoiser mismatch in PnP is limited (Liu et al., 2020; Shoushtari et al., 2022; Reehorst & Schniter, 2019; Laumont et al., 2022). Theoretical guarantees of RED with mismatched deep denoisers have been previously investigated for convex data-fidelity terms and nonexpansive denoisers (Shoushtari et al., 2022). A recent line of research has also used approximate MMSE denoisers in PnP (Reehorst & Schniter, 2019; Laumont et al., 2022).

**Domain Adaptation.** Distribution shift between training and inference datasets leads to mismatched DL models. Domain adaptation is commonly used to address distribution shift in DL. Domain adaptation has previously been used to address distribution shift in deep learning (Tommasi et al., 2012; 2013; Gopalan et al., 2011). Existing research in imaging problems focuses on adapting DL models from the source domain to the target domain by using the features extracted during inference for various problems such as image classification (Novi & Caputo, 2014), image reconstruction (Tirer & Giryes, 2019; Dou et al., 2019; Shocher et al., 2018), and image segmentation (Dou et al., 2019). In this work, we focus on scenarios where we have limited paired data from the target domain. Our domain adaptation fine-tunes pre-trained mismatched DL models using a small number of samples from the target domain, which is different from the inference-time domain adaptation.

**Our contributions.** *(1)* Our first contribution is a new theoretical analysis of PnP-ADMM accounting for the discrepancy between the desired and mismatched denoisers. Such analysis has not been considered in the prior work on PnP-ADMM. Our analysis is broadly applicable in the sense that it does *not* assume convex data-fidelity terms and nonexpansive denoisers. *(2)* Our second contribution is a comprehensive numerical study of distribution shifts in PnP through several well-known image datasets on the problem of image super-resolution. *(3)* Our third contribution is the illustration of simple data adaptation for addressing the problem of distribution shifts in PnP-ADMM. We show that one can successfully close the performance gap in PnP-ADMM due to distribution shifts by adapting the denoiser to the target distribution using very few samples.

---

**Algorithm 1** PnP-ADMM

---

1: **input:** $\boldsymbol{z}^0, \boldsymbol{s}^0 \in \mathbb{R}^n$, parameters $\sigma, \gamma > 0$.
2: **for** $k = 1, 2, 3, \cdots$ **do**
3:      $\boldsymbol{x}^k \leftarrow \mathsf{prox}_{\gamma g}(\boldsymbol{z}^{k-1} - \boldsymbol{s}^{k-1})$
4:      $\boldsymbol{z}^k \leftarrow \mathsf{D}_\sigma \left( \boldsymbol{x}^k + \boldsymbol{s}^{k-1} \right)$
5:      $\boldsymbol{s}^k \leftarrow \boldsymbol{s}^{k-1} + \boldsymbol{x}^k - \boldsymbol{z}^k$
6: **end for**

---

## 3    PROPOSED WORK

This section presents the convergence analysis of PnP-ADMM that accounts for the use of mismatched denoisers. It is worth noting that the theoretical analysis of PnP-ADMM has been previously discussed in (Chan, 2019; Teodoro et al., 2019; Ryu et al., 2019; Sun et al., 2021). The novelty of our work can be summarized in two aspects: (1) we analyze convergence with the mismatched priors; (2) our theory accommodates nonconvex $g$ and expansive denoisers.

### 3.1    PNP-ADMM WITH MISMATCHED DENOISER

We denote the target distribution as $p_{\boldsymbol{x}}$ and the mismatched distribution as $\widehat{p}_{\boldsymbol{x}}$. The mismatched denoiser $\widehat{\mathsf{D}}_\sigma$ is a *minimum mean squared error (MMSE)* estimator for the AWGN denoising problem

$$\boldsymbol{v} = \boldsymbol{x} + \boldsymbol{e} \quad \text{with} \quad \boldsymbol{x} \sim \widehat{p}_{\boldsymbol{x}}, \quad \boldsymbol{e} \sim \mathcal{N}(0, \sigma^2 \boldsymbol{I}). \tag{3}$$

The MMSE denoiser is the conditional mean estimator for (3) and can be expressed as

$$\widehat{\mathsf{D}}_\sigma(\boldsymbol{v}) \coloneqq \mathbb{E}[\boldsymbol{x}|\boldsymbol{v}] = \int_{\mathbb{R}^n} \boldsymbol{x} \widehat{p}_{\boldsymbol{x}|\boldsymbol{v}}(\boldsymbol{x}|\boldsymbol{v}) \, \mathrm{d}\boldsymbol{x}, \tag{4}$$

where $\widehat{p}_{\boldsymbol{x}|\boldsymbol{v}}(\boldsymbol{x}|\boldsymbol{v}) \propto G_\sigma(\boldsymbol{v} - \boldsymbol{x})\widehat{p}_{\boldsymbol{x}}(\boldsymbol{x})$, with $G_\sigma$ denoting the Gaussian density. We refer to the MMSE estimator $\widehat{\mathsf{D}}_\sigma$, corresponding to the mismatched data distribution $\widehat{p}_{\boldsymbol{x}}$, as the mismatched prior.

Since the integral (4) is generally intractable, in practice, the denoiser corresponds to a deep model trained to minimize the mean squared error (MSE) loss

$$\mathcal{L}(\widehat{\mathsf{D}}_\sigma) = \mathbb{E}\left[\|\boldsymbol{x} - \widehat{\mathsf{D}}_\sigma(\boldsymbol{v})\|_2^2\right]. \tag{5}$$

MMSE denoisers trained using the MSE loss are optimal with respect to the widely used image-quality metrics in denoising, such as signal-to-noise ratio (SNR), and have been extensively used in the PnP literature (Xu et al., 2020; Laumont et al., 2022; A. Bigdeli et al., 2017; Kadkhodaie & Simoncelli, 2021; Gan et al., 2023).

When using a mismatched prior in PnP-ADMM, we replace Step 4 in Algorithm 1 by

$$\boldsymbol{z}^k \leftarrow \widehat{\mathsf{D}}_\sigma \left( \boldsymbol{x}^k + \boldsymbol{s}^{k-1} \right), \tag{6}$$

where $\widehat{\mathsf{D}}_\sigma$ is the mismatched MMSE denoiser. To avoid confusion, we denote by $\boldsymbol{z}^k$ and $\overline{\boldsymbol{z}}^k$ the outputs of the mismatched and target denoisers at the $k$ iteration, respectively. Consequently, we have $\overline{\boldsymbol{z}}^k = \mathsf{D}_\sigma(\boldsymbol{x}^k + \boldsymbol{s}^{k-1})$, where $\mathsf{D}_\sigma$ is the target MMSE denoiser.

### 3.2    THEORETICAL ANALYSIS

Our analysis relies on the following set of assumptions that serve as sufficient conditions.

**Assumption 1.** *The prior distributions $p_{\boldsymbol{x}}$ and $\widehat{p}_{\boldsymbol{x}}$, denoted as target and mismatched priors respectively, are non-degenerate over $\mathbb{R}^n$.*

A distribution is considered degenerate over $\mathbb{R}^n$ if its support is confined to a lower-dimensional manifold than the dimensionality of $n$. Assumption 1 is useful to establish an explicit link between a

MMSE denoiser and its associated regularizer. For example, the regularizer $h$ associated with the target MMSE denoiser $\mathsf{D}_\sigma$ can be expressed as (see (Gribonval, 2011; Xu et al., 2020) for background)

$$h(\boldsymbol{x}) := \begin{cases} -\frac{1}{2\gamma}\|\boldsymbol{x} - \mathsf{D}_\sigma^{-1}(\boldsymbol{x})\|_2^2 + \frac{\sigma^2}{\gamma} h_\sigma(\mathsf{D}_\sigma^{-1}(\boldsymbol{x})) & \text{for } \boldsymbol{x} \in \mathsf{Im}(\mathsf{D}_\sigma) \\ +\infty & \text{for } \boldsymbol{x} \notin \mathsf{Im}(\mathsf{D}_\sigma), \end{cases} \tag{7}$$

where $\gamma > 0$ denotes the penalty parameter, $\mathsf{D}_\sigma^{-1} : \mathsf{Im}(\mathsf{D}_\sigma) \to \mathbb{R}^n$ represent a well defined and smooth inverse mapping over $\mathsf{Im}(\mathsf{D}_\sigma)$, and $h_\sigma(\cdot) := -\log(p_{\boldsymbol{u}}(\cdot))$, with $p_{\boldsymbol{u}}$ denoting the probability distribution over the AWGN corrupted observations

$$\boldsymbol{u} = \boldsymbol{x} + \boldsymbol{e} \quad \text{with} \quad \boldsymbol{x} \sim p_{\boldsymbol{x}}, \quad \boldsymbol{e} \sim \mathcal{N}(0, \sigma^2 \boldsymbol{I}),$$

(the derivation is provided in Section E.1 for completeness). Note that the smoothness of both $\mathsf{D}_\sigma^{-1}$ and $h_\sigma$ guarantees the smoothness of the function $h$. Additionally, similar connection exist between the mismatched MMSE denoiser $\widehat{\mathsf{D}}_\sigma$ and the regularizer $\hat{h}(\boldsymbol{x})$, with $\hat{h}_\sigma(\cdot) := -\log(\widehat{p}_{\boldsymbol{v}}(\cdot))$ characterizing the relationship between mismatched denoiser and shifted distribution.

**Assumption 2.** *The function $g$ is continuously differentiable.*

This assumption is a standard assumption used in nonconvex optimization, specifically in the context of inverse problems (Li & Li, 2018; Jiang et al., 2019; Yashtini, 2021).

**Assumption 3.** *The data-fidelity term and the implicit regularizers are bounded from below.*

Assumption 3 implies that there exists $f^* > -\infty$ such that $f(\boldsymbol{x}) \geq f^*$ for all $\boldsymbol{x} \in \mathbb{R}^n$.

**Assumption 4.** *The denoisers $\mathsf{D}_\sigma$ and $\widehat{\mathsf{D}}_\sigma$ have the same range $\mathsf{Im}(\mathsf{D}_\sigma)$. Additionally, functions $h$ and $\hat{h}$ associated with $\mathsf{D}_\sigma$ and $\widehat{\mathsf{D}}_\sigma$, are continuously differentiable with L-Lipschitz continuous gradients over $\mathsf{Im}(\mathsf{D}_\sigma)$.*

It is known (see (Gribonval, 2011; Xu et al., 2020)) that functions $h$ and $\hat{h}$ are infinitely differentiable over their ranges. The assumption that the two image denoisers have the same range is also a relatively mild assumption. Ideally, both denoisers would have the same range corresponding to the set of desired images. Assumption 4 is thus a mild extension that further requires Lipschitz continuity of the gradient over the range of denoisers.

**Assumption 5.** *The mismatched denoiser $\widehat{\mathsf{D}}_\sigma$ satisfies*

$$\|\widehat{\mathsf{D}}_\sigma(\boldsymbol{v}^k) - \mathsf{D}_\sigma(\boldsymbol{v}^k)\|_2 \leq \delta_k, \quad k = 1, 2, 3, \dots$$

*where $\widehat{\mathsf{D}}_\sigma$ is given in (4) and $\boldsymbol{v}^k = \boldsymbol{x}^k + \boldsymbol{s}^{k-1}$ in Algorithm 1.*

Our analysis assumes that at every iteration, PnP-ADMM uses a mismatched MMSE denoiser, derived from a shifted distribution. We consider the case where at iteration $k$ of PnP-ADMM, the distance of the outputs of $\mathsf{D}_\sigma$ and $\widehat{\mathsf{D}}_\sigma$ is bounded by a constant $\delta_k$.

**Assumption 6.** *For the sequence $\{\boldsymbol{x}^k, \boldsymbol{z}^k, \boldsymbol{s}^k\}$ generated by iterations of PnP-ADMM with mismatched MMSE denoiser in Algorithm 1, there exists a constant $R$ such that*

$$\left\|\boldsymbol{z}^k - \boldsymbol{z}^{k-1}\right\|_2 \leq R, \quad k = 1, 2, 3, \dots.$$

This assumption is a reasonable assumption since many images have bounded pixel values, for example $[0, 255]$ or $[0, 1]$.

We are now ready to present our convergence result under mismatched MMSE denoisers.

**Theorem 1.** *Run PnP-ADMM using a **mismatched** MMSE denoiser for $t \geq 1$ iterations under Assumptions 1-6 with the penalty parameter $0 < \gamma \leq 1/(4L)$. Then, we have*

$$\min_{1 \leq k \leq t} \left\|\nabla f\left(\boldsymbol{x}^k\right)\right\|_2^2 \leq \frac{1}{t} \sum_{k=1}^{t} \left\|\nabla f\left(\boldsymbol{x}^k\right)\right\|_2^2 \leq \frac{A_1}{t} + A_2 \bar{\varepsilon}_t$$

*where $A_1 > 0$ and $A_2 > 0$ are iteration independent constants and $\bar{\varepsilon}_t := (1/t)(\varepsilon_1 + \cdots + \varepsilon_t)$ is the error term that is an average of the quantities $\varepsilon_k := \max\{\delta_k, \delta_k^2\}$. In addition, if the sequence $\{\delta_i\}_{i \geq 1}$ is summable, $\|\nabla f(\boldsymbol{x}^t)\|_2 \to 0$ as $t \to \infty$.*

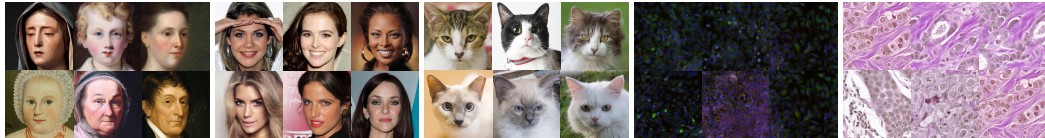

**Figure 2:** *Sample images from the datasets used for training the denoisers. From left to right: MetFaces (Karras et al., 2020), CelebA (Liu et al., 2015), AFHQ (Choi et al., 2020), RxRx1 (Sypetkowski et al., 2023), and BreCaHAD (Aksac et al., 2019).*

When we replace the mismatched MMSE denoiser with the *target* MMSE denoiser, we recover the traditional PnP-ADMM. To highlight the impact of the mismatch, we next provide the same statement but using the target denoiser.

**Theorem 2.** *Run PnP-ADMM with the target MMSE denoiser for $t \geq 1$ iterations under Assumptions 1-4 with the penalty parameter $0 < \gamma \leq 1/(4L)$. Then, we have*

$$\min_{1 \leq k \leq t} \left\| \nabla f\left(\boldsymbol{x}^k\right) \right\|_2^2 \leq \frac{1}{t} \sum_{k=1}^{t} \left\| \nabla f(\boldsymbol{x}^k) \right\|_2^2 \leq \frac{C}{t},$$

*where $C > 0$ is a constant independent of iteration.*

The proof of Theorem 1 is provided in the appendix. For completeness, we also provide the proof of Theorem 2. Our analysis relies on using the augmented Lagrangian $\phi$ as the Lyapunov function, where the augmented Lagrangian function value is decreasing and lower bounded for the sequence generated using Algorithm 1 (see (Wang et al., 2019) for additional discussion). Theorem 1 provides insight into the convergence of PnP-ADMM using mismatched MMSE denoisers. It states that if $\delta_k$ are summable, the iterates of PnP-ADMM with mismatched denoisers satisfy $\nabla f(\boldsymbol{x}^t) \rightarrow \boldsymbol{0}$ as $t \rightarrow \infty$. On the other hand, if the sequence $\delta_k$ is not summable, there is an error term that depends on the distance between the target and mismatched denoisers. Theorem 1 can be viewed as a more flexible alternative to the convergence analyses in (Sun et al., 2021; Chan, 2019; Ryu et al., 2019). While the analyses in the prior works assume convex $g$ and nonexpansive residual, nonexpansive or bounded denoisers, our analysis considers that denoiser $\mathsf{D}_\sigma$ is a mismatched MMSE estimator, where the mismatched denoiser distance to the target denoiser is bounded by $\delta_k$ at each iteration.

To summarize our theoretical results, PnP-ADMM using a mismatched MMSE denoiser approximates the solution obtained by PnP-ADMM using the target MMSE denoiser up to an error term that depends on the discrepancy between the denoisers. One can control The accuracy of PnP-ADMM using mismatched denoisers by controlling the error term $\overline{\varepsilon}_t$. This error term can be controlled by using domain adaptation techniques for decreasing the distance between mismatched and target denoisers, thus closing the gap in the performances of PnP-ADMM. We validate our theoretical analysis in Section 4 through numerical experiments, investigating the performance of PnP-ADMM under mismatched priors with varying levels of distribution shifts. Additionally, we use domain adaptation to illustrate the dependency of recovery errors in PnP-ADMM on the distance between mismatched and target denoisers. In domain adaptation, we fine-tune mismatched denoisers using a limited number of samples to minimize the distance between the mismatched and target distributions, consequently reducing errors in recovering solutions.

## 4 NUMERICAL VALIDATION

We consider PnP-ADMM with mismatched and adapted denoisers for the task of image super-resolution. Our first set of results shows how distribution shifts relate to the prior disparities and their impact on PnP recovery performance. Our second set of results shows the impact of domain adaptation on the denoiser gap and PnP performance. We use the traditional $l_2$-norm as the data-fidelity term. To provide an objective evaluation of the final image quality, we use two established quantitative metrics: Peak Signal-to-Noise Ratio (PSNR) and Structural Similarity Index (SSIM).

We use DRUNet architecture (Zhang et al., 2021) for all image denoisers. To model prior mismatch, we train denoisers on five image datasets: MetFaces (Karras et al., 2020), AFHQ (Choi et al., 2020), CelebA (Liu et al., 2015), BreCaHAD (Aksac et al., 2019), and RxRx1 (Sypetkowski et al., 2023).

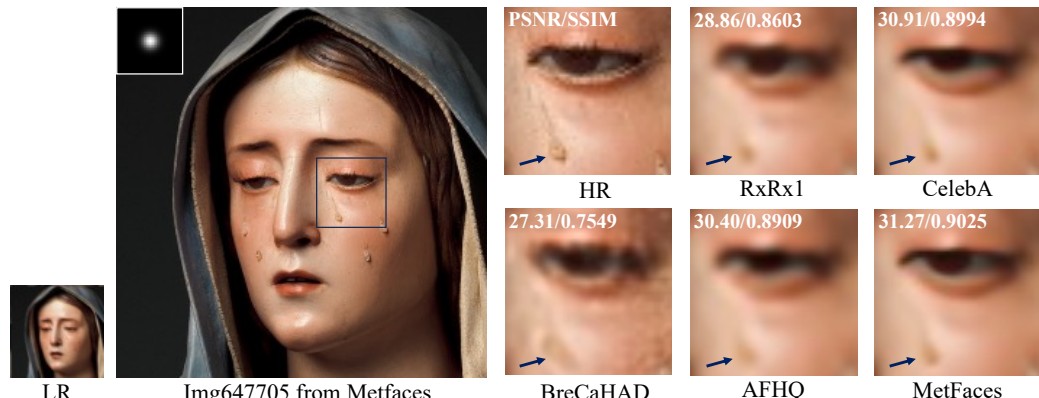

**Figure 3:** *Visual evaluation of PnP-ADMM on image super-resolution using denoisers trained on several datasets. Performance is reported in terms of PSNR (dB) and SSIM for an image from the MetFaces dataset. Images are downsampled by a scale of $s = 4$ and convolved with the blur kernel shown on the top left corner of the ground truth image. Note how the disparities in the training distributions of denoisers directly influence the performance of PnP. The denoisers containing images most similar to MetFaces offer the best performance.*

**Table 1:** *PSNR (dB) and SSIM values for image super-resolution using PnP-ADMM under different priors on a test set from the MetFaces (Karras et al., 2020). We highlighted the **best** performing and the worst performing priors. BreCaHAD is the worst prior that is also the one visually most different from MetFaces.*

| Kernels | Prior | $s = 2$ | | $s = 4$ | | Avg | |
|---|---|---|---|---|---|---|---|
| | | PSNR | SSIM | PSNR | SSIM | PSNR | SSIM |
| | BreCaHAD | 31.96 | 0.8108 | 28.41 | 0.6937 | 30.18 | 0.7522 |
| | RxRx1 | 33.45 | 0.8683 | 30.45 | 0.7906 | 31.95 | 0.8294 |
| | AFHQ | 33.74 | 0.8697 | 30.38 | 0.7825 | 32.06 | 0.8261 |
| | CelebA | 33.96 | 0.8731 | 30.62 | 0.7906 | 32.29 | 0.8318 |
| | MetFaces | **34.07** | **0.8755** | **31.15** | **0.8053** | **32.61** | **0.8404** |
| | BreCaHAD | 30.25 | 0.7489 | 28.99 | 0.7083 | 29.62 | 0.7286 |
| | RxRx1 | 32.22 | 0.8348 | 30.80 | 0.7948 | 31.51 | 0.8148 |
| | AFHQ | 32.63 | 0.8410 | 31.06 | 0.8014 | 31.84 | 0.8212 |
| | CelebA | 32.62 | 0.8404 | 31.30 | 0.8070 | 31.96 | 0.8237 |
| | MetFaces | **32.85** | **0.8457** | **31.44** | **0.8089** | **32.14** | **0.8273** |

Figure 2 illustrates samples from the datasets. Our training dataset consists of 1000 randomly chosen, resized, or cropped image slices, each measuring $256 \times 256$ pixels. Unlike several existing PnP methods (Sun et al., 2021; Liu et al., 2021) that suggest the inclusion of the spectral normalization layers into the CNN to enforce Lipschitz continuity on the denoisers, we directly train denoisers without any nonexpansiveness constraints.

## 4.1 IMPACT OF PRIOR MISMATCH

The observation model for single image super-resolution is $\boldsymbol{y} = \boldsymbol{SHx} + \boldsymbol{e}$, where $\boldsymbol{S} \in \mathbb{R}^{m \times n}$ is a standard $s$-fold downsampling matrix with $n = m \times s^2$, $\boldsymbol{H} \in \mathbb{R}^{n \times n}$ is a convolution with anti-aliasing kernel, and $\boldsymbol{e}$ is the noise. To compute the proximal map efficiently for the $l_2$-norm data-fidelity term (Step 3 in Algorithm 1), we use the closed-form solution outlined in (Zhang et al., 2021; Zhao et al., 2016). Similarly to (Zhang et al., 2021), we use four isotropic kernels with different standard deviation $\{0.7, 1.2, 1.6, 2\}$, as well as four anisotropic kernels depicted in Table 1. We perform downsampling at scales of $s = 2$ and $s = 4$.

Figure 3 illustrates the performance of PnP-ADMM using the target and four mismatched denoisers. Note the suboptimal performance of PnP-ADMM using mismatched denoisers trained on the BreCa-HAD, RxRx1, CelebA, and AFHQ datasets relative to PnP-ADMM using the target denoiser trained on the MetFaces dataset. Figure 3 illustrates how distribution shifts can lead to mismatched denoisers, subsequently impacting the performance of PnP-ADMM. It's worth noting that the denoiser trained

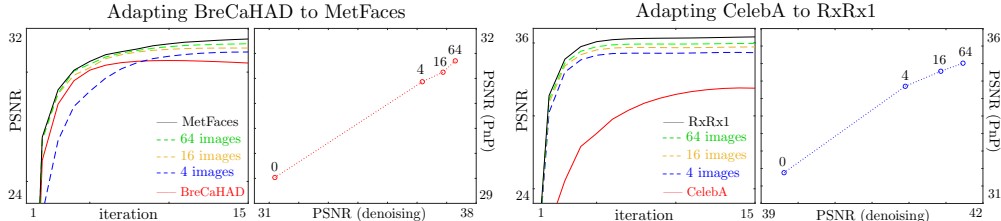

**Figure 4:** *Illustration of prior mismatch and adaption in PnP-ADMM, where a denoiser trained on one dataset (BreCaHAD (Aksac et al., 2019) or CelebA (Liu et al., 2015)) is used to recover an image from another dataset (MetFaces (Karras et al., 2020) or RxRx1 (Sypetkowski et al., 2023)). We plot the convergence of PnP-ADMM in terms of PSNR (first and third figures) and the influence of adapted denoisers on the performance of PnP-ADMM (second and fourth figures). Note how adaptation with even few samples is enough to nearly close the performance gap in PnP-ADMM.*

on the CelebA dataset (Liu et al., 2015), which consists of facial images similar to MetFaces, is the best-performing mismatched denoiser. Table 1 provides a quantitative evaluation of the PnP-ADMM performance with the target denoiser consistently outperforming all the other denoisers. Notably, the mismatched denoiser trained on the BreCaHAD dataset (Aksac et al., 2019), containing cell images that are most dissimilar to MetFaces, exhibits the worst performance.

## 4.2 DOMAIN ADAPTION

In domain adaptation, the pre-trained mismatched denoisers are updated using a limited number of data from the target distribution. We investigate two adaptation scenarios: in the first, we adapt the denoiser initially pre-trained on the BreCaHAD dataset to the MetFaces dataset, and in the second, we use the denoiser initially pre-trained on CelebA for adaptation to the RxRx1 dataset.

Figure 4 illustrates the influence of domain adaptation on denoising and PnP-ADMM. The reported results are tested on RxRx1 and MetFaces datasets for the super-resolution task. The kernel used is shown on the top left corner of the ground truth image in Figure 3 and the images are downsampled at the scale of $s = 4$. Note how the denoising performance improves as we increase the number of images used for domain adaptation. This indicates that domain adaptation reduces the distance of mismatched and target denoisers. Additionally, note the direct correlation between the denoising capabilities of priors and the performance of PnP-ADMM. Figure 4 shows that the performance of PnP-ADMM with mismatched denoisers can be significantly improved by adapting the mismatched denoiser to the target distribution, even with just four images from the target distribution.

Figure 5 presents visual examples illustrating domain adaptation in PnP-ADMM for image super-resolution. The recovery performance is shown for two test images from MetFaces using adapted denoisers against both target and mismatched denoisers. The experiment was conducted under the same settings as those in Figure 3. Note the effectiveness of domain adaptation in mitigating the impact of distribution shifts on PnP-ADMM. Table 2 provides quantitative results of several adapted priors on the test data. The results presented in Table 2 show the substantial impact of domain adaptation, using a limited number of data, in significantly narrowing the performance gap that emerges as a consequence of distribution shifts.

## 5 CONCLUSION

The work presented in this paper investigates the influence of using mismatched denoisers in PnP-ADMM, presents the corresponding theoretical analysis in terms of convergence, investigates the effect of mismatch on image super-resolution, and shows the ability of domain adaptation to reduce the effect of distribution mismatch. The theoretical results in this paper extend the recent PnP work by accommodating mismatched priors while eliminating the need for convex data-fidelity and nonexpansive denoiser assumptions. The empirical validation of PnP-ADMM involving mismatched priors and the domain adaptation strategy highlights the direct relationship between the gap in priors and the subsequent performance gap in the PnP-ADMM recovery, effectively reflecting the influence of distribution shifts on image priors.

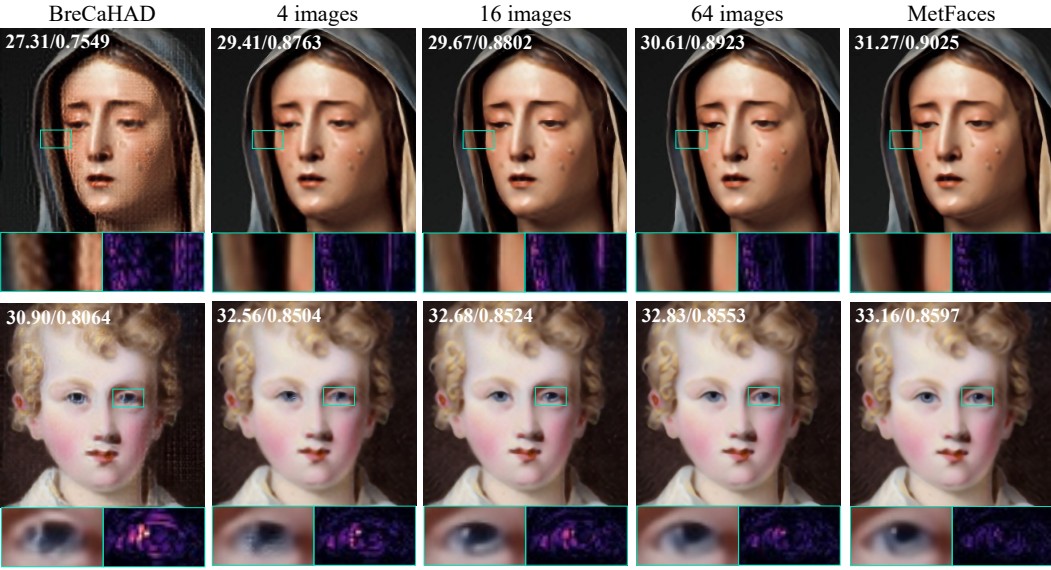

| BreCaHAD | 4 images | 16 images | 64 images | MetFaces |
|---|---|---|---|---|
| 27.31/0.7549 | 29.41/0.8763 | 29.67/0.8802 | 30.61/0.8923 | 31.27/0.9025 |
| 30.90/0.8064 | 32.56/0.8504 | 32.68/0.8524 | 32.83/0.8553 | 33.16/0.8597 |

**Figure 5:** *Visual comparison on super-resolution with target (MetFaces), mismatched (BreCaHAD), and adapted priors on two MetFaces test images. The images are downsampled by the scale of $s = 4$. The performance is reported in terms of PSNR (dB) and SSIM. Note how the recovery performance increases by adaptation of mismatched priors to a larger set of images from the target distribution.*

**Table 2:** *PSNR (dB) and SSIM comparison of super-resolution with mismatched, target, and adapted denoisers for the test set from MetFaces, averaged for indicated kernels. We highlighted the **target**, mismatched, and the best adapted priors.*

| Kernels | Prior | $s = 2$ | | $s = 4$ | | Avg | |
|---|---|---|---|---|---|---|---|
| | | PSNR | SSIM | PSNR | SSIM | PSNR | SSIM |
| | BreCaHAD | 31.96 | 0.8108 | 28.41 | 0.6937 | 30.18 | 0.7522 |
| | 4 imgs | 32.51 | 0.8510 | 30.57 | 0.7934 | 31.54 | 0.8222 |
| | 16 imgs | 33.10 | 0.8611 | 30.65 | 0.7961 | 31.89 | 0.8293 |
| | 32 imgs | 33.30 | 0.8649 | 30.81 | 0.8001 | 32.05 | 0.8325 |
| | 64 imgs | 33.59 | 0.8698 | 30.84 | 0.7994 | 32.21 | 0.8346 |
| | MetFaces | **34.07** | **0.8755** | **31.15** | **0.8053** | **32.61** | **0.8404** |
| | BreCaHAD | 30.25 | 0.7489 | 28.99 | 0.7083 | 29.62 | 0.7286 |
| | 4 imgs | 31.59 | 0.8215 | 30.86 | 0.7957 | 31.22 | 0.8086 |
| | 16 imgs | 32.19 | 0.8334 | 31.05 | 0.8009 | 31.62 | 0.8171 |
| | 32 imgs | 32.34 | 0.8371 | 31.18 | 0.8044 | 31.76 | 0.8207 |
| | 64 imgs | 32.47 | 0.8397 | 31.26 | 0.8059 | 31.86 | 0.8228 |
| | MetFaces | **32.85** | **0.8457** | **31.44** | **0.8089** | **32.14** | **0.8273** |

## LIMITATIONS

The basis of our analysis in this study relies on the assumption that the denoiser used in the inference accurately computes the MMSE denoiser for both target and mismatched distributions. While this assumption aligns well with deep denoisers trained via the MSE loss, it does not directly extend to denoisers trained using alternative loss functions, such as the $l_1$-norm or SSIM. As is customary in theoretical research, our analysis remains valid only under the fulfillment of these assumptions, which could potentially restrict its practical applicability. In our future work, we aim to enhance the results presented here by exploring new PnP strategies that can relax these convergence assumptions.

## REPRODUCIBILITY STATEMENT

We have provided the anonymous source code in the supplementary materials. The included README.md file contains detailed instructions on how to run the code and reproduce the results reported in the paper. The algorithm's pseudo-code is outlined in Algorithm 1, and for more comprehensive information about training, parameter selection, and other details, please refer to the Supplementary Section G. Additionally, for the theoretical findings presented in section 3, complete proofs, along with further clarifications on the assumptions and additional contextual information, can be found in the appendices A-E.

## ETHICS STATEMENT

To the best of our knowledge, this work does not give rise to any significant ethical concerns.

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

# Supplementary Material

## A    PROOF OF THEOREM 1

**Theorem.** *Run PnP-ADMM using a **mismatched** MMSE denoiser for $t \geq 1$ iterations under Assumptions 1-6 with the penalty parameter $0 < \gamma \leq 1/(4L)$. Then, we have*

$$\min_{1 \leq k \leq t} \left\| \nabla f\left(\boldsymbol{x}^k\right) \right\|_2^2 \leq \frac{1}{t} \sum_{k=1}^{t} \left\| \nabla f\left(\boldsymbol{x}^k\right) \right\|_2^2 \leq \frac{A_1}{t} + A_2 \overline{\varepsilon}_t$$

*where $A_1 > 0$ and $A_2 > 0$ are iteration independent constants and $\overline{\varepsilon}_t := (1/t)\left(\varepsilon_1 + \cdots + \varepsilon_t\right)$ is the error term that is an average of the quantities $\varepsilon_k := \max\{\delta_k, \delta_k^2\}$. In addition, if the sequence $\{\delta_i\}_{i \geq 1}$ is summable, $\|\nabla f(\boldsymbol{x}^t)\|_2 \to 0$ as $t \to \infty$.*

*Proof.* Note that from Lemma 1, we have

$$\left\| \boldsymbol{z}^k - \boldsymbol{z}^{k-1} \right\|_2^2 \leq \frac{2\gamma}{1 - 2\gamma L - 2\gamma^2 L^2} \left( \phi\left(\boldsymbol{x}^{k-1}, \boldsymbol{z}^{k-1}, \boldsymbol{s}^{k-1}\right) - \phi\left(\boldsymbol{x}^k, \boldsymbol{z}^k, \boldsymbol{s}^k\right) \right)$$

$$+ \frac{3}{4\left(1 - 2\gamma L - 2\gamma^2 L^2\right)} \delta_k^2 + \frac{2R}{1 - 2\gamma L - 2\gamma^2 L^2} \delta_k. \tag{8}$$

From the optimality conditions of the target MMSE denoiser $\mathsf{D}_\sigma$—where $\mathsf{D}_\sigma = \mathsf{prox}_{\gamma h}$ (see Section E.1)— and the proximal operator, for $\overline{\boldsymbol{z}}^k \in \mathsf{Im}(\mathsf{D}_\sigma)$, we have

$$\nabla h\left(\overline{\boldsymbol{z}}^k\right) + \frac{1}{\gamma}\left(\overline{\boldsymbol{z}}^k - \boldsymbol{x}^k - \boldsymbol{s}^{k-1}\right) = \boldsymbol{0} \quad \text{and} \quad \nabla g\left(\boldsymbol{x}^k\right) + \frac{1}{\gamma}\left(\boldsymbol{x}^k + \boldsymbol{s}^{k-1} - \boldsymbol{z}^{k-1}\right) = \boldsymbol{0}.$$

From this equation, for the objective function defined in (1), we can write

$$\left\| \nabla f\left(\boldsymbol{x}^k\right) \right\|_2 = \left\| \nabla g\left(\boldsymbol{x}^k\right) + \nabla h\left(\boldsymbol{x}^k\right) \right\|_2$$

$$= \left\| \nabla g\left(\boldsymbol{x}^k\right) + \frac{1}{\gamma}\left(\boldsymbol{x}^k + \boldsymbol{s}^{k-1} - \boldsymbol{z}^{k-1}\right) + \nabla h\left(\boldsymbol{x}^k\right) + \frac{1}{\gamma}\left(\boldsymbol{z}^{k-1} - \boldsymbol{x}^k - \boldsymbol{s}^{k-1}\right) \right\|_2$$

$$= \left\| \nabla h\left(\overline{\boldsymbol{z}}^k\right) + \frac{1}{\gamma}\left(\overline{\boldsymbol{z}}^k - \boldsymbol{x}^k - \boldsymbol{s}^{k-1}\right) + \nabla h\left(\boldsymbol{x}^k\right) - \nabla h\left(\overline{\boldsymbol{z}}^k\right) + \frac{1}{\gamma}\left(\boldsymbol{z}^{k-1} - \overline{\boldsymbol{z}}^k\right) \right\|_2$$

$$= \left\| \nabla h\left(\boldsymbol{x}^k\right) - \nabla h\left(\boldsymbol{z}^k\right) + \nabla h\left(\boldsymbol{z}^k\right) - \nabla h\left(\overline{\boldsymbol{z}}^k\right) + \frac{1}{\gamma}\left(\boldsymbol{z}^{k-1} - \overline{\boldsymbol{z}}^k\right) \right\|_2$$

$$\leq \left\| \nabla h\left(\boldsymbol{x}^k\right) - \nabla h\left(\boldsymbol{z}^k\right) \right\|_2 + \left\| \nabla h\left(\boldsymbol{z}^k\right) - \nabla h\left(\overline{\boldsymbol{z}}^k\right) \right\|_2 + \frac{1}{\gamma}\left\| \boldsymbol{z}^{k-1} - \overline{\boldsymbol{z}}^k \right\|_2$$

$$\leq L\left\| \boldsymbol{x}^k - \boldsymbol{z}^k \right\|_2 + L\left\| \boldsymbol{z}^k - \overline{\boldsymbol{z}}^k \right\|_2 + \frac{1}{\gamma}\left\| \boldsymbol{z}^{k-1} - \boldsymbol{z}^k \right\|_2 + \frac{1}{\gamma}\left\| \boldsymbol{z}^k - \overline{\boldsymbol{z}}^k \right\|_2$$

$$\leq \left( \frac{1}{\gamma} + \gamma L^2 \right) \left\| \boldsymbol{z}^k - \boldsymbol{z}^{k-1} \right\|_2 + \left( \frac{1}{\gamma} + L \right) \delta_k,$$

where we used triangle inequality in the first and second inequality. We also used

$$\left\| \boldsymbol{x}^k - \boldsymbol{z}^k \right\|_2 \leq \left\| \boldsymbol{s}^k - \boldsymbol{s}^{k-1} \right\|_2 = \gamma \left\| \nabla h(\boldsymbol{z}^k) - \nabla h(\boldsymbol{z}^{k-1}) \right\|_2 \leq \gamma L \left\| \boldsymbol{z}^k - \boldsymbol{z}^{k-1} \right\|_2$$

and Assumption 5 in the last inequality. By squaring both sides and using $(a+b)^2 \leq 2a^2 + 2b^2$, we get

$$\left\| \nabla f\left(\boldsymbol{x}^k\right) \right\|_2^2 \leq 2\left( \frac{1}{\gamma} + \gamma L^2 \right)^2 \left\| \boldsymbol{z}^k - \boldsymbol{z}^{k-1} \right\|_2^2 + 2\left( \frac{1}{\gamma} + L \right)^2 \delta_k^2.$$

By using the result from equation (8), we obtain

$$\left\| \nabla f\left(\boldsymbol{x}^k\right) \right\|_2^2 \leq \frac{4\left(1 + \gamma^2 L^2\right)^2}{\gamma\left(1 - 2\gamma L - 2\gamma^2 L^2\right)} \left( \phi\left(\boldsymbol{x}^{k-1}, \boldsymbol{z}^{k-1}, \boldsymbol{s}^{k-1}\right) - \phi\left(\boldsymbol{x}^k, \boldsymbol{z}^k, \boldsymbol{s}^k\right) \right)$$

$$+ \left( \frac{3\left(1 + \gamma^2 L^2\right)^2}{2\gamma^2\left(1 - 2\gamma L - 2\gamma^2 L^2\right)} + \frac{2\left(1 + \gamma L\right)^2}{\gamma^2} \right) \delta_k^2 + \frac{4R\left(1 + \gamma^2 L^2\right)^2}{\gamma^2\left(1 - 2\gamma L - 2\gamma^2 L^2\right)} \delta_k. \tag{9}$$

By averaging both sides of the bound over $t \geq 1$ and using the definition of error in $\varepsilon_k \coloneqq \max\{\delta_k, \delta_k^2\}$, we get

$$
\begin{aligned}
\min_{1 \leq k \leq t} \left\| \nabla f\left(\boldsymbol{x}^k\right) \right\|_2^2 &\leq \frac{1}{t} \sum_{k=1}^{t} \left\| \nabla f\left(\boldsymbol{x}^k\right) \right\|_2^2 \\
&\leq \frac{4\left(1+\gamma^2 L^2\right)^2 \left(\phi\left(\boldsymbol{x}^0, \boldsymbol{z}^0, \boldsymbol{s}^0\right) - \phi\left(\boldsymbol{x}^t, \boldsymbol{z}^t, \boldsymbol{s}^t\right)\right)}{\gamma\left(1 - 2\gamma L - 2\gamma^2 L^2\right)} \frac{1}{t} + A_2 \bar{\varepsilon}_t \\
&\leq \frac{4\left(1+\gamma^2 L^2\right)^2 \left(\phi\left(\boldsymbol{x}^0, \boldsymbol{z}^0, \boldsymbol{s}^0\right) - \phi^*\right)}{\gamma\left(1 - 2\gamma L - 2\gamma^2 L^2\right)} \frac{1}{t} + A_2 \bar{\varepsilon}_t \\
&\leq \frac{A_1}{t} + A_2 \bar{\varepsilon}_t,
\end{aligned}
\tag{10}
$$

where $\bar{\varepsilon}_t \coloneqq (1/t)\left(\varepsilon_1 + \cdots + \varepsilon_t\right)$,

$$
A_1 \coloneqq 4\left(1+\gamma^2 L^2\right)^2 \left(\phi(\boldsymbol{x}^0, \boldsymbol{z}^0, \boldsymbol{s}^0) - \phi^*\right) / \left(\gamma\left(1 - 2\gamma L - 2\gamma^2 L^2\right)\right),
$$
$$
A_2 \coloneqq \left((3 + 16R)\left(1 + \gamma^2 L^2\right) / \left(2\gamma^2\left(1 - 2\gamma L - 2\gamma^2 L^2\right)\right) + 2\left(1/\gamma + L\right)^2\right),
$$

and we used the fact that $\phi^* \leq \phi\left(\boldsymbol{x}^t, \boldsymbol{z}^t, \boldsymbol{s}^t\right)$ from Lemma 2. Note that we used the following inequality to get the result in equation (10)

$$
\begin{aligned}
&\left(\frac{3\left(1+\gamma^2 L^2\right)^2}{2\gamma^2\left(1 - 2\gamma L - 2\gamma^2 L^2\right)} + 2\left(\frac{1}{\gamma} + L\right)^2\right) \delta_k^2 + \frac{4R\left(1+\gamma^2 L^2\right)^2}{\gamma^2\left(1 - 2\gamma L - 2\gamma^2 L^2\right)} \delta_k \\
&\leq \max\{\delta_k, \delta_k^2\} \left(\frac{3\left(1+\gamma^2 L^2\right)^2}{2\gamma^2\left(1 - 2\gamma L - 2\gamma^2 L^2\right)} + 2\left(\frac{1}{\gamma} + L\right)^2 + \frac{4R\left(1+\gamma^2 L^2\right)^2}{\gamma^2\left(1 - 2\gamma L - 2\gamma^2 L^2\right)}\right) \\
&= \left(\frac{(3 + 8R)\left(1+\gamma^2 L^2\right)^2}{2\gamma^2\left(1 - 2\gamma L - 2\gamma^2 L^2\right)} + 2\left(\frac{1}{\gamma} + L\right)^2\right) \varepsilon_k.
\end{aligned}
$$

Note that if the sequence of distances of denoisers $\{\delta_i\}_{i \geq 1}$ is summable, then $\{\varepsilon_i = \max\{\delta_i, \delta_i^2\}\}_{i \geq 1}$ is also be summable. Consequently, $\|\nabla f(\boldsymbol{x}^t)\|_2 \to 0$ as $t \to \infty$. □

**Remark 1.** Note that by using (8) when the sequence $\{\delta_i\}_{i \geq 1}$ is summable, we have

$$
\frac{1}{t} \sum_{k=1}^{t} \left\| \boldsymbol{z}^k - \boldsymbol{z}^{k-1} \right\|_2^2 \leq 0 \quad \text{as} \quad t \to \infty,
\tag{11}
$$

which ensures that $\|\boldsymbol{z}^k - \boldsymbol{z}^{k-1}\|_2 \to 0$ as $k \to \infty$. Since

$$
\left\| \boldsymbol{s}^k - \boldsymbol{s}^{k-1} \right\|_2 \leq \gamma L \left\| \boldsymbol{z}^k - \boldsymbol{z}^{k-1} \right\|_2 \quad \text{and} \quad \left\| \boldsymbol{s}^k - \boldsymbol{s}^{k-1} \right\|_2 = \left\| \boldsymbol{x}^k - \boldsymbol{z}^k \right\|_2,
\tag{12}
$$

we conclude that $\|\boldsymbol{x}^k - \boldsymbol{x}^{k-1}\|_2 \to 0$ and $\|\boldsymbol{s}^k - \boldsymbol{s}^{k-1}\|_2 \to 0$ as $k \to \infty$.

# B  USEFUL RESULTS FOR THEOREM 1

**Lemma 1.** *Assume that Assumptions 1-6 hold and let the sequence $\{\boldsymbol{x}^k, \boldsymbol{z}^k, \boldsymbol{s}^k\}$ be generated via iterations of PnP-ADMM with **mismatched** MMSE denoiser using the penalty parameter $0 < \gamma \leq 1/(4L)$. Then for the augmented Lagrangian defined in (2), we have that*

$$
\phi\left(\boldsymbol{x}^k, \boldsymbol{z}^k, \boldsymbol{s}^k\right) \leq \phi\left(\boldsymbol{x}^{k-1}, \boldsymbol{z}^{k-1}, \boldsymbol{s}^{k-1}\right) - \left(\frac{1 - 2\gamma L - 2\gamma^2 L^2}{2\gamma}\right) \left\| \boldsymbol{z}^k - \boldsymbol{z}^{k-1} \right\|_2^2 + \frac{3}{8\gamma} \delta_k^2 + \frac{R}{\gamma} \delta_k.
$$

*where $R$ is defined in Assumption 6.*

*Proof.* From the smoothness of $\hat{h}$ for any $\boldsymbol{z}^k \in \mathsf{Im}(\mathsf{D}_\sigma)$ in Assumption 4, the optimality condition for the mismatched MMSE denoiser, and the Lagrange multiplier update rule in the form of $\boldsymbol{s}^k = \boldsymbol{s}^{k-1} + \boldsymbol{x}^k - \boldsymbol{z}^k$, we have

$$\nabla \hat{h}\left(\boldsymbol{z}^k\right) = \frac{1}{\gamma}\left(\boldsymbol{s}^{k-1} + \boldsymbol{x}^k - \boldsymbol{z}^k\right) = \frac{1}{\gamma}\boldsymbol{s}^k$$

and

$$\left\|\boldsymbol{s}^k - \boldsymbol{s}^{k-1}\right\|_2 = \left\|\gamma \nabla \hat{h}\left(\boldsymbol{z}^k\right) - \gamma \nabla \hat{h}\left(\boldsymbol{z}^{k-1}\right)\right\|_2 \leq \gamma L \left\|\boldsymbol{z}^k - \boldsymbol{z}^{k-1}\right\|_2, \tag{13}$$

where we used $L-$Lipschitz continuity of $\nabla \hat{h}$ from Assumption (4) in the last inequality. From this equation and the Lagrange multiplier update rule, we have

$$\phi\left(\boldsymbol{x}^k, \boldsymbol{z}^k, \boldsymbol{s}^k\right) - \phi\left(\boldsymbol{x}^k, \boldsymbol{z}^k, \boldsymbol{s}^{k-1}\right) = \frac{1}{\gamma}\left(\boldsymbol{s}^k - \boldsymbol{s}^{k-1}\right)^\mathsf{T}\left(\boldsymbol{x}^k - \boldsymbol{z}^k\right) = \frac{1}{\gamma}\left\|\boldsymbol{s}^k - \boldsymbol{s}^{k-1}\right\|_2^2$$

$$\leq \gamma L^2 \left\|\boldsymbol{z}^k - \boldsymbol{z}^{k-1}\right\|_2^2. \tag{14}$$

From the fact that $h$ (regularizer associated with target MMSE denoiser $\mathsf{D}_\sigma$) has a $L-$Lipschitz continuous gradient over the set $\mathsf{Im}(\mathsf{D}_\sigma)$ (Assumption 4), we have

$$h\left(\overline{\boldsymbol{z}}^k\right) - h\left(\boldsymbol{z}^{k-1}\right) \leq \nabla h\left(\overline{\boldsymbol{z}}^k\right)^\mathsf{T}\left(\overline{\boldsymbol{z}}^k - \boldsymbol{z}^{k-1}\right) + \frac{L}{2}\left\|\overline{\boldsymbol{z}}^k - \boldsymbol{z}^{k-1}\right\|_2^2, \tag{15}$$

where $\overline{\boldsymbol{z}}^k = \mathsf{D}_\sigma(\boldsymbol{x}^k + \boldsymbol{s}^{k-1})$. From the smoothness of $h$ for any $\overline{\boldsymbol{z}}^k \in \mathsf{Im}(\mathsf{D}_\sigma)$, the optimality condition for mismatched MMSE denoiser ($\overline{\boldsymbol{z}}^k = \mathsf{D}_\sigma(\boldsymbol{x}^k + \boldsymbol{s}^{k-1}) = \mathsf{prox}_{\gamma h}(\boldsymbol{x}^k + \boldsymbol{s}^{k-1})$, see derivation in Section E.1), and the Lagrange multiplier update rule $\boldsymbol{s}^k = \boldsymbol{s}^{k-1} + \boldsymbol{x}^k - \boldsymbol{z}^k$, we have

$$\nabla h\left(\overline{\boldsymbol{z}}^k\right) + \frac{1}{\gamma}\left(\overline{\boldsymbol{z}}^k - \boldsymbol{x}^k - \boldsymbol{s}^{k-1}\right) = \mathbf{0},$$

which implies that

$$\nabla h\left(\overline{\boldsymbol{z}}^k\right) = \frac{1}{\gamma}\left(\boldsymbol{x}^k + \boldsymbol{s}^{k-1} - \overline{\boldsymbol{z}}^k\right) = \frac{1}{\gamma}\boldsymbol{s}^k + \frac{1}{\gamma}\left(\boldsymbol{z}^k - \overline{\boldsymbol{z}}^k\right). \tag{16}$$

By combining equations (15) and (16), we obtain

$$h\left(\overline{\boldsymbol{z}}^k\right) - h\left(\boldsymbol{z}^{k-1}\right) \leq \frac{1}{\gamma}\left(\boldsymbol{s}^k\right)^\mathsf{T}\left(\overline{\boldsymbol{z}}^k - \boldsymbol{z}^{k-1}\right) + \frac{1}{\gamma}\left(\boldsymbol{z}^k - \overline{\boldsymbol{z}}^k\right)^\mathsf{T}\left(\overline{\boldsymbol{z}}^k - \boldsymbol{z}^{k-1}\right) + \frac{L}{2}\left\|\overline{\boldsymbol{z}}^k - \boldsymbol{z}^{k-1}\right\|_2^2. \tag{17}$$

For the target MMSE denoiser $\mathsf{D}_\sigma$, we know that $\overline{\boldsymbol{z}}^k \in \mathsf{Im}(\mathsf{D}_\sigma)$ minimizes

$$\psi_{\gamma h}\left(\boldsymbol{z}\right) := \frac{1}{2\gamma}\left\|\boldsymbol{z} - \left(\boldsymbol{x}^k + \boldsymbol{s}^{k-1}\right)\right\|_2^2 + h\left(\boldsymbol{z}\right). \tag{18}$$

From Assumption 4, we know that $\nabla h$ is $L-$Lipschitz continuous over $\mathsf{Im}(\mathsf{D}_\sigma)$, which implies

$$\left\|\nabla \psi_{\gamma h}\left(\boldsymbol{u}\right) - \nabla \psi_{\gamma h}\left(\boldsymbol{v}\right)\right\|_2 \leq \left(\frac{1}{\gamma} + L\right)\left\|\boldsymbol{u} - \boldsymbol{v}\right\|_2 \quad \forall \boldsymbol{u}, \boldsymbol{v} \in \mathsf{Im}(\mathsf{D}_\sigma).$$

From the smoothness of $\psi_{\gamma h}$ and the fact that $\overline{\boldsymbol{z}}^k$ minimizes it, we have

$$\psi_{\gamma h}\left(\boldsymbol{z}^k\right) \leq \psi_{\gamma h}\left(\overline{\boldsymbol{z}}^k\right) + \nabla \psi_{\gamma h}\left(\overline{\boldsymbol{z}}^k\right)^\mathsf{T}\left(\boldsymbol{z}^k - \overline{\boldsymbol{z}}^k\right) + \left(\frac{1}{2\gamma} + \frac{L}{2}\right)\left\|\boldsymbol{z}^k - \overline{\boldsymbol{z}}^k\right\|_2^2$$

$$= \psi_{\gamma h}\left(\overline{\boldsymbol{z}}^k\right) + \left(\frac{1}{2\gamma} + \frac{L}{2}\right)\left\|\boldsymbol{z}^k - \overline{\boldsymbol{z}}^k\right\|_2^2.$$

By using the definition of function $\psi_{\gamma h}$ in (18), the Lagrange multiplier update rule $\boldsymbol{s}^k = \boldsymbol{s}^{k-1} + \boldsymbol{x}^k - \boldsymbol{z}^k$ and rearranging the terms, we obtain

$$h\left(\boldsymbol{z}^k\right) - h\left(\overline{\boldsymbol{z}}^k\right) \leq \frac{1}{2\gamma}\left\|\overline{\boldsymbol{z}}^k - \left(\boldsymbol{x}^k + \boldsymbol{s}^{k-1}\right)\right\|_2^2 - \frac{1}{2\gamma}\left\|\boldsymbol{z}^k - \left(\boldsymbol{x}^k + \boldsymbol{s}^{k-1}\right)\right\|_2^2 + \left(\frac{1}{2\gamma} + \frac{L}{2}\right)\left\|\boldsymbol{z}^k - \overline{\boldsymbol{z}}^k\right\|_2^2$$

$$= \frac{1}{2\gamma}\left(\overline{\boldsymbol{z}}^k + \boldsymbol{z}^k - 2\left(\boldsymbol{x}^k + \boldsymbol{s}^{k-1}\right)\right)^\mathsf{T}\left(\overline{\boldsymbol{z}}^k - \boldsymbol{z}^k\right) + \left(\frac{1}{2\gamma} + \frac{L}{2}\right)\left\|\boldsymbol{z}^k - \overline{\boldsymbol{z}}^k\right\|_2^2$$

$$= \frac{1}{\gamma}\left(\boldsymbol{s}^k\right)^\mathsf{T}\left(\boldsymbol{z}^k - \overline{\boldsymbol{z}}^k\right) + \frac{1}{2\gamma}\left\|\boldsymbol{z}^k - \overline{\boldsymbol{z}}^k\right\|_2^2 + \left(\frac{1}{2\gamma} + \frac{L}{2}\right)\left\|\boldsymbol{z}^k - \overline{\boldsymbol{z}}^k\right\|_2^2$$

$$= \frac{1}{\gamma}\left(\boldsymbol{s}^k\right)^\mathsf{T}\left(\boldsymbol{z}^k - \overline{\boldsymbol{z}}^k\right) + \left(\frac{1}{\gamma} + \frac{L}{2}\right)\left\|\boldsymbol{z}^k - \overline{\boldsymbol{z}}^k\right\|_2^2. \tag{19}$$

Now for the augmented Lagrangian, we have

$$
\phi\left(\boldsymbol{x}^{k}, \boldsymbol{z}^{k}, \boldsymbol{s}^{k-1}\right)-\phi\left(\boldsymbol{x}^{k}, \boldsymbol{z}^{k-1}, \boldsymbol{s}^{k-1}\right)=h\left(\boldsymbol{z}^{k}\right)-h\left(\boldsymbol{z}^{k-1}\right)+\frac{1}{\gamma}\left(\boldsymbol{s}^{k-1}\right)^{\top}\left(\boldsymbol{z}^{k-1}-\boldsymbol{z}^{k}\right)
$$
$$
+\frac{1}{2\gamma}\left(2\boldsymbol{x}^{k}-\boldsymbol{z}^{k}-\boldsymbol{z}^{k-1}\right)^{\top}\left(\boldsymbol{z}^{k-1}-\boldsymbol{z}^{k}\right)
$$
$$
=h\left(\boldsymbol{z}^{k}\right)-h\left(\boldsymbol{z}^{k-1}\right)+\frac{1}{\gamma}\left(\boldsymbol{s}^{k-1}\right)^{\top}\left(\boldsymbol{z}^{k-1}-\boldsymbol{z}^{k}\right)
$$
$$
+\frac{1}{\gamma}\left(\boldsymbol{s}^{k}-\boldsymbol{s}^{k-1}\right)^{\top}\left(\boldsymbol{z}^{k-1}-\boldsymbol{z}^{k}\right)-\frac{1}{2\gamma}\left\|\boldsymbol{z}^{k}-\boldsymbol{z}^{k-1}\right\|_{2}^{2}
$$
$$
=h\left(\boldsymbol{z}^{k}\right)-h\left(\overline{\boldsymbol{z}}^{k}\right)+h\left(\overline{\boldsymbol{z}}^{k}\right)-h\left(\boldsymbol{z}^{k-1}\right)
$$
$$
+\frac{1}{\gamma}\left(\boldsymbol{s}^{k}\right)^{\top}\left(\boldsymbol{z}^{k-1}-\boldsymbol{z}^{k}\right)-\frac{1}{2\gamma}\left\|\boldsymbol{z}^{k}-\boldsymbol{z}^{k-1}\right\|_{2}^{2}, \quad (20)
$$

where we used the Lagrange multiplier update rule in the second equality. By plugging (17) and (19) into (20) and rearranging the terms, we obtain

$$
\phi\left(\boldsymbol{x}^{k}, \boldsymbol{z}^{k}, \boldsymbol{s}^{k-1}\right)-\phi\left(\boldsymbol{x}^{k}, \boldsymbol{z}^{k-1}, \boldsymbol{s}^{k-1}\right)\leq\frac{1}{\gamma}\left(\boldsymbol{z}^{k}-\overline{\boldsymbol{z}}^{k}\right)^{\top}\left(\overline{\boldsymbol{z}}^{k}-\boldsymbol{z}^{k-1}\right)+\frac{L}{2}\left\|\overline{\boldsymbol{z}}^{k}-\boldsymbol{z}^{k-1}\right\|_{2}^{2}
$$
$$
-\frac{1}{2\gamma}\left\|\boldsymbol{z}^{k}-\boldsymbol{z}^{k-1}\right\|_{2}^{2}+\left(\frac{1}{\gamma}+\frac{L}{2}\right)\left\|\boldsymbol{z}^{k}-\overline{\boldsymbol{z}}^{k}\right\|_{2}^{2}
$$
$$
=\frac{1}{\gamma}\left(\boldsymbol{z}^{k}-\overline{\boldsymbol{z}}^{k}\right)^{\top}\left(\overline{\boldsymbol{z}}^{k}-\boldsymbol{z}^{k}+\boldsymbol{z}^{k}-\boldsymbol{z}^{k-1}\right)+\frac{L}{2}\left\|\overline{\boldsymbol{z}}^{k}-\boldsymbol{z}^{k}+\boldsymbol{z}^{k}-\boldsymbol{z}^{k-1}\right\|_{2}^{2}
$$
$$
-\frac{1}{2\gamma}\left\|\boldsymbol{z}^{k}-\boldsymbol{z}^{k-1}\right\|_{2}^{2}+\left(\frac{1}{\gamma}+\frac{L}{2}\right)\left\|\boldsymbol{z}^{k}-\overline{\boldsymbol{z}}^{k}\right\|_{2}^{2}. \quad (21)
$$

By using $\|\boldsymbol{a}+\boldsymbol{b}\|^{2}\leq 2\|\boldsymbol{a}\|^{2}+2\|\boldsymbol{b}\|^{2}$, we can write

$$
\phi\left(\boldsymbol{x}^{k}, \boldsymbol{z}^{k}, \boldsymbol{s}^{k-1}\right)-\phi\left(\boldsymbol{x}^{k}, \boldsymbol{z}^{k-1}, \boldsymbol{s}^{k-1}\right)\leq\frac{1}{\gamma}\left(\boldsymbol{z}^{k}-\overline{\boldsymbol{z}}^{k}\right)^{\top}\left(\overline{\boldsymbol{z}}^{k}-\boldsymbol{z}^{k}\right)+\frac{1}{\gamma}\left(\boldsymbol{z}^{k}-\overline{\boldsymbol{z}}^{k}\right)^{\top}\left(\boldsymbol{z}^{k}-\boldsymbol{z}^{k-1}\right)
$$
$$
+L\left\|\overline{\boldsymbol{z}}^{k}-\boldsymbol{z}^{k}\right\|_{2}^{2}+L\left\|\boldsymbol{z}^{k}-\boldsymbol{z}^{k-1}\right\|_{2}^{2}-\frac{1}{2\gamma}\left\|\boldsymbol{z}^{k}-\boldsymbol{z}^{k-1}\right\|_{2}^{2}+\left(\frac{1}{\gamma}+\frac{L}{2}\right)\left\|\boldsymbol{z}^{k}-\overline{\boldsymbol{z}}^{k}\right\|_{2}^{2}
$$
$$
\leq-\frac{1}{\gamma}\left\|\boldsymbol{z}^{k}-\overline{\boldsymbol{z}}^{k}\right\|_{2}^{2}+\frac{1}{\gamma}\left(\boldsymbol{z}^{k}-\overline{\boldsymbol{z}}^{k}\right)^{\top}\left(\boldsymbol{z}^{k}-\boldsymbol{z}^{k-1}\right)-\left(\frac{1-2\gamma L}{2\gamma}\right)\left\|\boldsymbol{z}^{k}-\boldsymbol{z}^{k-1}\right\|_{2}^{2}
$$
$$
+\frac{1}{2\gamma}\left\|\boldsymbol{z}^{k}-\overline{\boldsymbol{z}}^{k}\right\|_{2}^{2}+\frac{11}{8\gamma}\left\|\boldsymbol{z}^{k}-\overline{\boldsymbol{z}}^{k}\right\|_{2}^{2}
$$
$$
=\frac{1}{\gamma}\left(\boldsymbol{z}^{k}-\overline{\boldsymbol{z}}^{k}\right)^{\top}\left(\boldsymbol{z}^{k}-\boldsymbol{z}^{k-1}\right)-\left(\frac{1-2\gamma L}{2\gamma}\right)\left\|\boldsymbol{z}^{k}-\boldsymbol{z}^{k-1}\right\|_{2}^{2}+\frac{3}{8\gamma}\left\|\boldsymbol{z}^{k}-\overline{\boldsymbol{z}}^{k}\right\|_{2}^{2}, \quad (22)
$$

where we used the fact that $0<\gamma\leq 1/(4L)$ in the second inequality. From Assumption 6, we have

$$
\left\|\boldsymbol{z}^{k}-\boldsymbol{z}^{k-1}\right\|_{2}\leq R. \quad (23)
$$

Using this equation, Assumption 6, and the bound on denoiser distance in Assumption 5, we obtain

$$
\phi\left(\boldsymbol{x}^{k}, \boldsymbol{z}^{k}, \boldsymbol{s}^{k-1}\right)\leq\phi\left(\boldsymbol{x}^{k}, \boldsymbol{z}^{k-1}, \boldsymbol{s}^{k-1}\right)+\frac{1}{\gamma}\left(\boldsymbol{z}^{k}-\overline{\boldsymbol{z}}^{k}\right)^{\top}\left(\boldsymbol{z}^{k}-\boldsymbol{z}^{k-1}\right)
$$
$$
-\left(\frac{1-2\gamma L}{2\gamma}\right)\left\|\boldsymbol{z}^{k}-\boldsymbol{z}^{k-1}\right\|_{2}^{2}+\frac{3}{8\gamma}\left\|\boldsymbol{z}^{k}-\overline{\boldsymbol{z}}^{k}\right\|_{2}^{2}
$$
$$
\leq\phi\left(\boldsymbol{x}^{k}, \boldsymbol{z}^{k-1}, \boldsymbol{s}^{k-1}\right)-\left(\frac{1-2\gamma L}{2\gamma}\right)\left\|\boldsymbol{z}^{k}-\boldsymbol{z}^{k-1}\right\|_{2}^{2}
$$
$$
+\frac{1}{\gamma}\left\|\boldsymbol{z}^{k}-\overline{\boldsymbol{z}}^{k}\right\|_{2}\left\|\boldsymbol{z}^{k}-\boldsymbol{z}^{k-1}\right\|_{2}+\frac{3}{8\gamma}\left\|\boldsymbol{z}^{k}-\overline{\boldsymbol{z}}^{k}\right\|_{2}^{2}
$$
$$
\leq\phi\left(\boldsymbol{x}^{k}, \boldsymbol{z}^{k-1}, \boldsymbol{s}^{k-1}\right)-\left(\frac{1-2\gamma L}{2\gamma}\right)\left\|\boldsymbol{z}^{k}-\boldsymbol{z}^{k-1}\right\|_{2}^{2}+\frac{3}{8\gamma}\delta_{k}^{2}+\frac{R}{\gamma}\delta_{k}. \quad (24)
$$

Note that from $\boldsymbol{x}^k = \mathsf{prox}_{\gamma g}(\boldsymbol{z}^{k-1} - \boldsymbol{s}^{k-1})$, we have

$$\frac{1}{2\gamma} \left\| \boldsymbol{x}^k - \boldsymbol{z}^{k-1} + \boldsymbol{s}^{k-1} \right\|_2^2 + g\left(\boldsymbol{x}^k\right) = \min_{\boldsymbol{x} \in \mathbb{R}^n} \left\{ \frac{1}{2\gamma} \left\| \boldsymbol{x} - \boldsymbol{z}^{k-1} + \boldsymbol{s}^{k-1} \right\|_2^2 + g\left(\boldsymbol{x}\right) \right\}$$

$$\leq \frac{1}{2\gamma} \left\| \boldsymbol{x}^{k-1} - \boldsymbol{z}^{k-1} + \boldsymbol{s}^{k-1} \right\|_2^2 + g\left(\boldsymbol{x}^{k-1}\right),$$

which implies that

$$\phi\left(\boldsymbol{x}^k, \boldsymbol{z}^{k-1}, \boldsymbol{s}^{k-1}\right) \leq \phi\left(\boldsymbol{x}^{k-1}, \boldsymbol{z}^{k-1}, \boldsymbol{s}^{k-1}\right). \tag{25}$$

By combining equations (14), (24) and (25), we obtain

$$\phi\left(\boldsymbol{x}^k, \boldsymbol{z}^k, \boldsymbol{s}^k\right) \leq \phi\left(\boldsymbol{x}^{k-1}, \boldsymbol{z}^{k-1}, \boldsymbol{s}^{k-1}\right) - \left(\frac{1 - 2\gamma L - 2\gamma^2 L^2}{2\gamma}\right) \left\| \boldsymbol{z}^k - \boldsymbol{z}^{k-1} \right\|_2^2 + \frac{3}{8\gamma}\delta_k^2 + \frac{R}{\gamma}\delta_k.$$

$$\square$$

**Lemma 2.** *Assume that Assumptions 1-6 hold and let the sequence $\{\boldsymbol{x}^k, \boldsymbol{z}^k, \boldsymbol{s}^k\}$ be generated via PnP-ADMM with **mismatched** MMSE denoiser using penalty parameter $0 < \gamma \leq 1/(4L)$. Then, the augment Lagrangian $\phi$ defined in (2) is bounded from below*

$$\inf_{k \geq 0} \phi\left(\boldsymbol{x}^k, \boldsymbol{z}^k, \boldsymbol{s}^k\right) \geq \phi^* > -\infty.$$

*Proof.* From the smoothness of $h$ (regularizer associated with the target denoiser $\mathsf{D}_\sigma$) for any $\overline{\boldsymbol{z}}^k \in \mathsf{Im}(\mathsf{D}_\sigma)$, the optimality condition for the target MMSE denoiser, and the Lagrange multiplier update rule in the form of $\boldsymbol{s}^k = \boldsymbol{s}^{k-1} + \boldsymbol{x}^k - \boldsymbol{z}^k$, we have

$$\nabla h\left(\overline{\boldsymbol{z}}^k\right) = \frac{1}{\gamma}\left(\boldsymbol{s}^{k-1} + \boldsymbol{x}^k - \overline{\boldsymbol{z}}^k\right) = \frac{1}{\gamma}\boldsymbol{s}^k + \frac{1}{\gamma}\left(\boldsymbol{z}^k - \overline{\boldsymbol{z}}^k\right). \tag{26}$$

From the Lipschitz continuity of $\nabla h$ in Assumption 4 and the fact that $\gamma \leq 1/(4L) < 1/L$, we have

$$h\left(\boldsymbol{x}^k\right) \leq h\left(\boldsymbol{z}^k\right) + \nabla h\left(\boldsymbol{z}^k\right)^\mathsf{T}\left(\boldsymbol{x}^k - \boldsymbol{z}^k\right) + \frac{L}{2}\left\|\boldsymbol{x}^k - \boldsymbol{z}^k\right\|_2^2$$

$$< h\left(\boldsymbol{z}^k\right) + \nabla h\left(\boldsymbol{z}^k\right)^\mathsf{T}\left(\boldsymbol{x}^k - \boldsymbol{z}^k\right) + \frac{1}{2\gamma}\left\|\boldsymbol{x}^k - \boldsymbol{z}^k\right\|_2^2.$$

By using this inequality and equation (26), we can write

$$\phi\left(\boldsymbol{x}^k, \boldsymbol{z}^k, \boldsymbol{s}^k\right) = g\left(\boldsymbol{x}^k\right) + h\left(\boldsymbol{z}^k\right) + \frac{1}{\gamma}\left(\boldsymbol{s}^k\right)^\mathsf{T}\left(\boldsymbol{x}^k - \boldsymbol{z}^k\right) + \frac{1}{2\gamma}\left\|\boldsymbol{x}^k - \boldsymbol{z}^k\right\|_2^2$$

$$= g\left(\boldsymbol{x}^k\right) + h\left(\boldsymbol{z}^k\right) + \nabla h\left(\overline{\boldsymbol{z}}^k\right)^\mathsf{T}\left(\boldsymbol{x}^k - \boldsymbol{z}^k\right) + \frac{1}{\gamma}\left(\overline{\boldsymbol{z}}^k - \boldsymbol{z}^k\right)^\mathsf{T}\left(\boldsymbol{x}^k - \boldsymbol{z}^k\right)$$

$$+ \frac{1}{2\gamma}\left\|\boldsymbol{x}^k - \boldsymbol{z}^k\right\|_2^2$$

$$= g\left(\boldsymbol{x}^k\right) + h\left(\boldsymbol{z}^k\right) + \nabla h\left(\boldsymbol{z}^k\right)^\mathsf{T}\left(\boldsymbol{x}^k - \boldsymbol{z}^k\right) + \frac{1}{2\gamma}\left\|\boldsymbol{x}^k - \boldsymbol{z}^k\right\|_2^2$$

$$+ \left(\nabla h\left(\overline{\boldsymbol{z}}^k\right) - \nabla h\left(\boldsymbol{z}^k\right)\right)^\mathsf{T}\left(\boldsymbol{x}^k - \boldsymbol{z}^k\right) + \frac{1}{\gamma}\left(\overline{\boldsymbol{z}}^k - \boldsymbol{z}^k\right)^\mathsf{T}\left(\boldsymbol{x}^k - \boldsymbol{z}^k\right)$$

$$> g\left(\boldsymbol{x}^k\right) + h\left(\boldsymbol{x}^k\right) - \left\|\nabla h\left(\overline{\boldsymbol{z}}^k\right) - \nabla h\left(\boldsymbol{z}^k\right)\right\|_2 \left\|\boldsymbol{x}^k - \boldsymbol{z}^k\right\|_2$$

$$- \frac{1}{\gamma}\left\|\overline{\boldsymbol{z}}^k - \boldsymbol{z}^k\right\|_2 \left\|\boldsymbol{x}^k - \boldsymbol{z}^k\right\|_2, \tag{27}$$

where we added and subtracted the term $\nabla h(\boldsymbol{z}^k)^\mathsf{T}(\boldsymbol{x}^k - \boldsymbol{z}^k)$ in the third line and used Cauchy-Schwarz inequality in the last line.

From the Lagrange multiplier update rule $\boldsymbol{s}^k = \boldsymbol{s}^{k-1} + \boldsymbol{x}^k - \boldsymbol{z}^k$, equations (13) and (23), we obtain

$$\left\|\boldsymbol{x}^k - \boldsymbol{z}^k\right\|_2 = \left\|\boldsymbol{s}^k - \boldsymbol{s}^{k-1}\right\|_2 \leq \gamma L \left\|\boldsymbol{z}^k - \boldsymbol{z}^{k-1}\right\|_2 \leq \gamma L R. \tag{28}$$

By using the bound on the distance of target and mismatched denoisers in Assumption 5, Lipschitz continuity of $\nabla h$ in Assumption 4, equations (27) and (28), we get

$$\phi\left(\boldsymbol{x}^k, \boldsymbol{z}^k, \boldsymbol{s}^k\right) > g\left(\boldsymbol{x}^k\right) + h\left(\boldsymbol{x}^k\right) - (1+\gamma L)\, RL\delta_k. \tag{29}$$

From the fact that both functions $g$ and $h$ are bounded from below in Assumption 3 and the fact that $\gamma$, $\delta_k$, $R$, and $L$ are finite constants, we conclude that the augmented Lagrangian is bounded from below. For the special case of $k = 0$, we have $\|\overline{\boldsymbol{z}}^0 - \boldsymbol{z}^0\|_2 = 0$, as both $\overline{\boldsymbol{z}}^0$ and $\boldsymbol{z}^0$ are initialization of the algorithm with target and mismatched denoisers and can be deliberately set to be identical. Consequently, from equation (27) we have

$$\phi\left(\boldsymbol{x}^0, \boldsymbol{z}^0, \boldsymbol{s}^0\right)$$
$$> g\left(\boldsymbol{x}^0\right) + h\left(\boldsymbol{x}^0\right) - \left\|\nabla h\left(\overline{\boldsymbol{z}}^0\right) - \nabla h\left(\boldsymbol{z}^0\right)\right\|_2 \left\|\boldsymbol{x}^0 - \boldsymbol{z}^0\right\|_2 - \frac{1}{\gamma} \left\|\overline{\boldsymbol{z}}^0 - \boldsymbol{z}^0\right\|_2 \left\|\boldsymbol{x}^0 - \boldsymbol{z}^0\right\|_2$$
$$\geq g\left(\boldsymbol{x}^0\right) + h\left(\boldsymbol{x}^0\right) - (L + \frac{1}{\gamma}) \left\|\overline{\boldsymbol{z}}^0 - \boldsymbol{z}^0\right\|_2 \left\|\boldsymbol{x}^0 - \boldsymbol{z}^0\right\|_2$$
$$= f(\boldsymbol{x}^0)$$
$$\geq f^*,$$

where we used Assumption 3. This equation and equation (29) imply the existence of $\phi^* = \phi(\boldsymbol{x}^*, \boldsymbol{z}^*, \boldsymbol{s}^*) > -\infty$ such that we have almost surely $\phi^* \leq \phi(\boldsymbol{x}^k, \boldsymbol{z}^k, \boldsymbol{s}^k)$, for all $k \geq 0$. $\qquad\square$

## C  PROOF OF THEOREM 2

**Theorem.** *Run PnP-ADMM with the target MMSE denoiser for $t \geq 1$ iterations under Assumptions 1-4 with the penalty parameter $0 < \gamma \leq 1/(4L)$. Then, we have*

$$\min_{1 \leq k \leq t} \left\|\nabla f\left(\boldsymbol{x}^k\right)\right\|_2^2 \leq \frac{1}{t}\sum_{k=1}^{t}\left\|\nabla f(\boldsymbol{x}^k)\right\|_2^2 \leq \frac{C}{t},$$

*where $C > 0$ is a constant independent of iteration.*

*Proof.* Note that for PnP-ADMM with the MMSE denoiser, Lemma 3 states

$$\left\|\boldsymbol{z}^k - \boldsymbol{z}^{k-1}\right\|_2^2 \leq \frac{2\gamma}{1 - \gamma L - 2\gamma^2 L^2}\left(\phi\left(\boldsymbol{x}^{k-1}, \boldsymbol{z}^{k-1}, \boldsymbol{s}^{k-1}\right) - \phi\left(\boldsymbol{x}^k, \boldsymbol{z}^k, \boldsymbol{s}^k\right)\right). \tag{30}$$

By averaging over $t \geq 1$ iterations and using the fact that the augmented Lagrangian is bounded from below in Lemma 4, we obtain

$$\frac{1}{t}\sum_{k=1}^{t}\left\|\boldsymbol{z}^k - \boldsymbol{z}^{k-1}\right\| \leq \frac{B}{t}\left(\phi\left(\boldsymbol{x}^0, \boldsymbol{z}^0, \boldsymbol{s}^0\right) - \phi\left(\boldsymbol{x}^t, \boldsymbol{z}^t, \boldsymbol{s}^t\right)\right)$$
$$\leq \frac{B}{t}\left(\phi\left(\boldsymbol{x}^0, \boldsymbol{z}^0, \boldsymbol{s}^0\right) - \phi^*\right), \tag{31}$$

where $B \coloneqq 2\gamma(\phi(\boldsymbol{x}^0, \boldsymbol{z}^0, \boldsymbol{s}^0) - \phi^*)/(1 - \gamma L - 2\gamma^2 L^2)$. Note that since $\phi^*$ is the infimum defined in Lemma 4 and $0 < \gamma \leq 1/(4L)$, $B$ is a positive constant. From the optimality conditions for the MMSE denoiser $\mathsf{D}_\sigma$ and $\boldsymbol{x}^k = \mathsf{prox}_{\gamma g}(\boldsymbol{z}^{k-1} - \boldsymbol{s}^{k-1})$, we have

$$\nabla g\left(\boldsymbol{x}^k\right) + \frac{1}{\gamma}\left(\boldsymbol{x}^k + \boldsymbol{s}^{k-1} - \boldsymbol{z}^{k-1}\right) = \boldsymbol{0} \quad\text{and}\quad \nabla h\left(\boldsymbol{z}^k\right) + \frac{1}{\gamma}\left(\boldsymbol{z}^k - \boldsymbol{x}^k - \boldsymbol{s}^{k-1}\right) = \boldsymbol{0}. \tag{32}$$

By using the $L$-Lipschitz continuity of $\nabla h$ and the Lagrange multiplier update rule in the form of $\boldsymbol{s}^k = \boldsymbol{s}^{k-1} + \boldsymbol{z}^k - \boldsymbol{x}^k$, we can write

$$\left\|\nabla h\left(\boldsymbol{x}^k\right) - \nabla h\left(\boldsymbol{z}^k\right)\right\|_2 \leq L\left\|\boldsymbol{x}^k - \boldsymbol{z}^k\right\|_2 = L\left\|\boldsymbol{s}^k - \boldsymbol{s}^{k-1}\right\|_2$$
$$= \gamma L\left\|\nabla h\left(\boldsymbol{z}^k\right) - \nabla h\left(\boldsymbol{z}^{k-1}\right)\right\|_2$$
$$\leq \gamma L^2\left\|\boldsymbol{z}^k - \boldsymbol{z}^{k-1}\right\|_2.$$

By using this equation and equation (32), we have for the objective function in (1)

$$
\begin{aligned}
\left\| \nabla f\left(\boldsymbol{x}^k\right) \right\|_2 &= \left\| \nabla g\left(\boldsymbol{x}^k\right) + \nabla h\left(\boldsymbol{x}^k\right) \right\|_2 \\
&= \left\| \nabla g\left(\boldsymbol{x}^k\right) + \frac{1}{\gamma}\left(\boldsymbol{x}^k + \boldsymbol{s}^{k-1} - \boldsymbol{z}^{k-1}\right) + \nabla h\left(\boldsymbol{x}^k\right) + \frac{1}{\gamma}\left(\boldsymbol{z}^{k-1} - \boldsymbol{s}^{k-1} - \boldsymbol{x}^k\right) \right\|_2 \\
&= \| \nabla h\left(\boldsymbol{z}^k\right) + \frac{1}{\gamma}\left(\boldsymbol{z}^k - \boldsymbol{x}^k - \boldsymbol{s}^{k-1}\right) + \nabla h\left(\boldsymbol{x}^k\right) - \nabla h\left(\boldsymbol{z}^k\right) + \frac{1}{\gamma}\left(\boldsymbol{z}^{k-1} - \boldsymbol{z}^k\right) \|_2 \\
&\leq \left\| \nabla h\left(\boldsymbol{x}^k\right) - \nabla h\left(\boldsymbol{z}^k\right) \right\|_2 + \frac{1}{\gamma}\left\| \boldsymbol{z}^k - \boldsymbol{z}^{k-1} \right\|_2 \\
&\leq \left(\frac{1}{\gamma} + \gamma L^2\right)\left\| \boldsymbol{z}^k - \boldsymbol{z}^{k-1} \right\|_2
\end{aligned}
$$

where we used triangle inequality in the first inequality. By squaring both sides, averaging over $t \geq 1$ iterations, and usi equation (31), we get the desired result

$$
\min_{1 \leq k \leq t} \left\| \nabla f\left(\boldsymbol{x}^k\right) \right\|_2^2 \leq \frac{1}{t}\sum_{k=1}^t \left\| \nabla f\left(\boldsymbol{x}^k\right) \right\|_2^2 \leq \frac{C}{t}
$$

where $C \coloneqq B(1 + \gamma^2 L^2)/\gamma^2$ is a positive constant. $\qquad\square$

## D  Useful results for Theorem 2

**Lemma 3.** *Assume that Assumptions 1-4 hold and let the sequence $\{\boldsymbol{x}^k, \boldsymbol{z}^k, \boldsymbol{s}^k\}$ be generated via iterations of PnP-ADMM with the MMSE denoiser using the penalty parameter $0 < \gamma < 1/(4L)$. Then for the augmented Lagrangian defined in 2, we have that*

$$
\phi\left(\boldsymbol{x}^k, \boldsymbol{z}^k, \boldsymbol{s}^k\right) \leq \phi\left(\boldsymbol{x}^{k-1}, \boldsymbol{z}^{k-1}, \boldsymbol{s}^{k-1}\right) - \left(\frac{1 - \gamma L - 2\gamma^2 L^2}{2\gamma}\right)\left\| \boldsymbol{z}^k - \boldsymbol{z}^{k-1} \right\|_2^2 .
$$

*Proof.* From the smoothness of $h$ for any $\boldsymbol{z}^k \in \mathsf{Im}(\mathsf{D}_\sigma)$, the optimality condition for the MMSE denoiser, and the Lagrange multiplier update rule in the form of $\boldsymbol{s}^k = \boldsymbol{s}^{k-1} + \boldsymbol{x}^k - \boldsymbol{z}^k$, we have

$$
\nabla h\left(\boldsymbol{z}^k\right) = \frac{1}{\gamma}\left(\boldsymbol{x}^k + \boldsymbol{s}^{k-1} - \boldsymbol{z}^k\right) = \frac{1}{\gamma}\boldsymbol{s}^k .
$$

From this equality and the definition of the augmented Lagrangian in (2), we have

$$
\begin{aligned}
\phi\left(\boldsymbol{x}^k, \boldsymbol{z}^k, \boldsymbol{s}^k\right) - \phi\left(\boldsymbol{x}^k, \boldsymbol{z}^k, \boldsymbol{s}^{k-1}\right) &= \frac{1}{\gamma}\left(\boldsymbol{s}^k - \boldsymbol{s}^{k-1}\right)^{\mathsf{T}}\left(\boldsymbol{x}^k - \boldsymbol{z}^k\right) \\
&= \frac{1}{\gamma}\left\| \boldsymbol{s}^k - \boldsymbol{s}^{k-1} \right\|_2^2 = \gamma\left\| \nabla h\left(\boldsymbol{z}^k\right) - \nabla h\left(\boldsymbol{z}^{k-1}\right) \right\|_2^2 \\
&\leq \gamma L^2\left\| \boldsymbol{z}^k - \boldsymbol{z}^{k-1} \right\|_2^2 ,
\end{aligned}
\tag{33}
$$

where in the last line we used $L$-Lipschitz continuity of $\nabla h$ in Assumption 4. Additionally, we have

$$
\begin{aligned}
h\left(\boldsymbol{z}^k\right) - h\left(\boldsymbol{z}^{k-1}\right) &\leq \nabla h\left(\boldsymbol{z}^k\right)^{\mathsf{T}}\left(\boldsymbol{z}^k - \boldsymbol{z}^{k-1}\right) + \frac{L}{2}\left\| \boldsymbol{z}^k - \boldsymbol{z}^{k-1} \right\|_2^2 \\
&= \frac{1}{\gamma}\left(\boldsymbol{s}^k\right)^{\mathsf{T}}\left(\boldsymbol{z}^k - \boldsymbol{z}^{k-1}\right) + \frac{L}{2}\left\| \boldsymbol{z}^k - \boldsymbol{z}^{k-1} \right\|_2^2 .
\end{aligned}
$$

Now by using this equation and the definition of the augmented Lagrangian, we have

$$
\phi\left(\boldsymbol{x}^{k}, \boldsymbol{z}^{k}, \boldsymbol{s}^{k-1}\right) - \phi\left(\boldsymbol{x}^{k}, \boldsymbol{z}^{k-1}, \boldsymbol{s}^{k-1}\right) = h\left(\boldsymbol{z}^{k}\right) - h\left(\boldsymbol{z}^{k-1}\right) + \frac{1}{\gamma}\left(\boldsymbol{s}^{k-1}\right)^{\mathsf{T}}\left(\boldsymbol{z}^{k-1} - \boldsymbol{z}^{k}\right)
$$

$$
+ \frac{1}{2\gamma}\left\|\boldsymbol{x}^{k} - \boldsymbol{z}^{k}\right\|_{2}^{2} - \frac{1}{2\gamma}\left\|\boldsymbol{x}^{k} - \boldsymbol{z}^{k-1}\right\|_{2}^{2}
$$

$$
= h\left(\boldsymbol{z}^{k}\right) - h\left(\boldsymbol{z}^{k-1}\right) + \frac{1}{\gamma}\left(\boldsymbol{s}^{k-1}\right)^{\mathsf{T}}\left(\boldsymbol{z}^{k-1} - \boldsymbol{z}^{k}\right)
$$

$$
+ \frac{1}{2\gamma}\left(2\boldsymbol{x}^{k} - \boldsymbol{z}^{k} - \boldsymbol{z}^{k-1}\right)^{\mathsf{T}}\left(\boldsymbol{z}^{k-1} - \boldsymbol{z}^{k}\right)
$$

$$
= h\left(\boldsymbol{z}^{k}\right) - h\left(\boldsymbol{z}^{k-1}\right) + \frac{1}{\gamma}\left(\boldsymbol{s}^{k-1}\right)^{\mathsf{T}}\left(\boldsymbol{z}^{k-1} - \boldsymbol{z}^{k}\right)
$$

$$
+ \frac{1}{\gamma}\left(\boldsymbol{s}^{k} - \boldsymbol{s}^{k-1}\right)^{\mathsf{T}}\left(\boldsymbol{z}^{k-1} - \boldsymbol{z}^{k}\right) - \frac{1}{2\gamma}\left\|\boldsymbol{z}^{k} - \boldsymbol{z}^{k-1}\right\|_{2}^{2}
$$

$$
\leq \frac{1}{\gamma}\left(\boldsymbol{s}^{k}\right)^{\mathsf{T}}\left(\boldsymbol{z}^{k} - \boldsymbol{z}^{k-1}\right) + \frac{L}{2}\left\|\boldsymbol{z}^{k} - \boldsymbol{z}^{k-1}\right\|_{2}^{2} + \frac{1}{\gamma}\left(\boldsymbol{s}^{k-1}\right)^{\mathsf{T}}\left(\boldsymbol{z}^{k-1} - \boldsymbol{z}^{k}\right)
$$

$$
+ \frac{1}{\gamma}\left(\boldsymbol{s}^{k} - \boldsymbol{s}^{k-1}\right)^{\mathsf{T}}\left(\boldsymbol{z}^{k-1} - \boldsymbol{z}^{k}\right) - \frac{1}{2\gamma}\left\|\boldsymbol{z}^{k} - \boldsymbol{z}^{k-1}\right\|_{2}^{2}
$$

$$
\leq -\left(\frac{1 - \gamma L}{2\gamma}\right)\left\|\boldsymbol{z}^{k} - \boldsymbol{z}^{k-1}\right\|_{2}^{2}. \tag{34}
$$

Note that from $\boldsymbol{x}^{k} = \mathsf{prox}_{\gamma g}(\boldsymbol{z}^{k-1} - \boldsymbol{s}^{k-1})$, we have

$$
\frac{1}{2\gamma}\left\|\boldsymbol{x}^{k} - \boldsymbol{z}^{k-1} + \boldsymbol{s}^{k-1}\right\|_{2}^{2} + g\left(\boldsymbol{x}^{k}\right) = \min_{\boldsymbol{x}\in\mathbb{R}^{n}}\left\{\frac{1}{2\gamma}\left\|\boldsymbol{x} - \boldsymbol{z}^{k-1} + \boldsymbol{s}^{k-1}\right\|_{2}^{2} + g\left(\boldsymbol{x}\right)\right\}
$$

$$
\leq \frac{1}{2\gamma}\left\|\boldsymbol{x}^{k-1} - \boldsymbol{z}^{k-1} + \boldsymbol{s}^{k-1}\right\|_{2}^{2} + g\left(\boldsymbol{x}^{k-1}\right),
$$

which implies that

$$
\phi\left(\boldsymbol{x}^{k}, \boldsymbol{z}^{k-1}, \boldsymbol{s}^{k-1}\right) \leq \phi\left(\boldsymbol{x}^{k-1}, \boldsymbol{z}^{k-1}, \boldsymbol{s}^{k-1}\right). \tag{35}
$$

Now by combining the results from equations (33), (34) and (35), we have

$$
\phi\left(\boldsymbol{x}^{k}, \boldsymbol{z}^{k}, \boldsymbol{s}^{k}\right) \leq \phi\left(\boldsymbol{x}^{k-1}, \boldsymbol{z}^{k-1}, \boldsymbol{s}^{k-1}\right) - \left(\frac{1 - \gamma L - 2\gamma^{2}L^{2}}{2\gamma}\right)\left\|\boldsymbol{z}^{k} - \boldsymbol{z}^{k-1}\right\|_{2}^{2}.
$$

$\square$

**Lemma 4.** *Assume that Assumptions 1-4 hold and let the sequence $\{\boldsymbol{x}^{k}, \boldsymbol{z}^{k}, \boldsymbol{s}^{k}\}$ be generated via PnP-ADMM with the MMSE denoiser using the penalty parameter $0 < \gamma < 1/(4L)$. Then the augmented Lagrangian $\phi$ defined in (2) is bounded from below*

$$
\inf_{k\geq 0}\phi\left(\boldsymbol{x}^{k}, \boldsymbol{z}^{k}, \boldsymbol{s}^{k}\right) \geq \phi^{*} > -\infty.
$$

*Proof.* From the smoothness of $h$ for any $\boldsymbol{z}^{k} \in \mathsf{Im}(\mathsf{D}_{\sigma})$, the optimality condition for the MMSE denoiser, and the Lagrange multiplier update rule in the form of $\boldsymbol{s}^{k} = \boldsymbol{s}^{k-1} + \boldsymbol{x}^{k} - \boldsymbol{z}^{k}$, we have

$$
\nabla h\left(\boldsymbol{z}^{k}\right) = \frac{1}{\gamma}\left(\boldsymbol{x}^{k} + \boldsymbol{s}^{k-1} - \boldsymbol{z}^{k}\right) = \frac{1}{\gamma}\boldsymbol{s}^{k}. \tag{36}
$$

By using the $L$-Lipschitz continuity of $\nabla h$ in Assumption 4, we have that

$$
h\left(\boldsymbol{x}^{k}\right) \leq h\left(\boldsymbol{z}^{k}\right) + \nabla h\left(\boldsymbol{z}^{k}\right)^{\mathsf{T}}\left(\boldsymbol{x}^{k} - \boldsymbol{z}^{k}\right) + \frac{L}{2}\left\|\boldsymbol{x}^{k} - \boldsymbol{z}^{k}\right\|_{2}^{2}. \tag{37}
$$

From equations (36), (37) and the fact that $\gamma L < 1$, we have

$$
\phi\left(\boldsymbol{x}^{k}, \boldsymbol{z}^{k}, \boldsymbol{s}^{k}\right) = g\left(\boldsymbol{x}^{k}\right) + h\left(\boldsymbol{z}^{k}\right) + \frac{1}{\gamma}\left(\boldsymbol{s}^{k}\right)^{\mathsf{T}}\left(\boldsymbol{x}^{k} - \boldsymbol{z}^{k}\right) + \frac{1}{2\gamma}\left\|\boldsymbol{x}^{k} - \boldsymbol{z}^{k}\right\|_{2}^{2}
$$

$$
> g\left(\boldsymbol{x}^{k}\right) + h\left(\boldsymbol{z}^{k}\right) + \nabla h\left(\boldsymbol{z}^{k}\right)^{\mathsf{T}}\left(\boldsymbol{x}^{k} - \boldsymbol{z}^{k}\right) + \frac{L}{2}\left\|\boldsymbol{x}^{k} - \boldsymbol{z}^{k}\right\|_{2}^{2}
$$

$$
> g\left(\boldsymbol{x}^{k}\right) + h\left(\boldsymbol{x}^{k}\right).
$$

Note that since both functions $g$ and $h$ are bounded from below from Assumption 3, we conclude that the augmented Lagrangian is bounded from below. This implies that there exists $-\infty < \phi^* \leq \phi(\boldsymbol{x}^k, \boldsymbol{z}^k, \boldsymbol{s}^k)$, for all $k \geq 0$.  □

## E  BACKGROUND MATERIAL

### E.1  MMSE DENOISING AS PROXIMAL OPERATOR

The connection between MMSE estimation and regularized inversion was established by Gribonval in (Gribonval, 2011), and this relationship has been explored in various contexts (Gribonval & Machart, 2013; Kazerouni et al., 2013; Gribonval & Nikolova, 2021; Gan et al., 2023). This connection was formally linked to Plug-and-Play (PnP) methods in (Xu et al., 2020), resulting in a novel interpretation of MMSE denoisers within the framework of PnP. In this section, we investigate the fundamental argument that bridges MMSE denoising and proximal operators.

The MMSE estimator for the following AWGN denoising problem

$$\boldsymbol{u} = \boldsymbol{x} + \boldsymbol{e} \quad \text{with} \quad \boldsymbol{x} \sim \widehat{p}_{\boldsymbol{x}}, \quad \boldsymbol{e} \sim \mathcal{N}(0, \sigma^2 \boldsymbol{I}), \tag{38}$$

is expressed as

$$\mathsf{D}_\sigma(\boldsymbol{u}) := \mathbb{E}[\boldsymbol{x}|\boldsymbol{u}] = \int_{\mathbb{R}^n} \boldsymbol{x} p_{\boldsymbol{x}|\boldsymbol{u}}(\boldsymbol{x}|\boldsymbol{u}) \, \mathsf{d}\boldsymbol{x}. \tag{39}$$

From *Tweedie's formula*, we can express the estimator (39) as

$$\mathsf{D}_\sigma(\boldsymbol{u}) = \boldsymbol{u} - \sigma^2 \nabla h_\sigma(\boldsymbol{u}) \quad \text{with} \quad h_\sigma(\boldsymbol{u}) = -\log(p_{\boldsymbol{u}}(\boldsymbol{u})), \tag{40}$$

which is derived by differentiating (39) using the expression for the probability distribution given by

$$p_{\boldsymbol{u}}(\boldsymbol{u}) = (p_{\boldsymbol{x}} * \phi_\sigma)(\boldsymbol{u}) = \int_{\mathbb{R}^n} \phi_\sigma(\boldsymbol{u} - \boldsymbol{x}) p_{\boldsymbol{x}}(\boldsymbol{x}) \, \mathsf{d}\boldsymbol{x}, \tag{41}$$

where

$$\phi_\sigma(\boldsymbol{x}) := \frac{1}{(2\pi\sigma^2)^{\frac{n}{2}}} \exp\left(-\frac{\|\boldsymbol{x}\|^2}{2\sigma^2}\right).$$

Since $\phi_\sigma$ is infinitely differentiable, the same applies to $p_{\boldsymbol{u}}$ and $\mathsf{D}_\sigma$. As demonstrated in Lemma 2 of (Gribonval, 2011), the Jacobian of $\mathsf{D}_\sigma$ is positive definite:

$$\mathsf{JD}_\sigma(\boldsymbol{u}) = \boldsymbol{I} - \sigma^2 \mathsf{H}h_\sigma(\boldsymbol{u}) \succ 0, \quad \boldsymbol{u} \in \mathbb{R}^n, \tag{42}$$

where $\mathsf{H}h_\sigma$ represents the Hessian matrix of the function $h_\sigma$. Additionally, Assumption 1 implies that $\mathsf{D}_\sigma$ is a *one-to-one* mapping from $\mathbb{R}^n$ to $\mathsf{Im}(\mathsf{D}_\sigma)$. This implies that $(\mathsf{D}_\sigma)^{-1} : \mathsf{Im}(\mathsf{D}_\sigma) \to \mathbb{R}^n$ is well defined and infinitely differentiable over $\mathsf{Im}(\mathsf{D}_\sigma)$, as outlined in Lemma 1 of (Gribonval, 2011). Consequently, this indicates that the regularizer $h$ in (7) is also infinitely differentiable for any $\boldsymbol{x} \in \mathsf{Im}(\mathsf{D}_\sigma)$.

We will now establish that

$$\mathsf{D}_\sigma(\boldsymbol{u}) = \mathsf{prox}_{\gamma h}(\boldsymbol{u}) = \underset{\boldsymbol{x} \in \mathbb{R}^n}{\arg\min} \left\{ \frac{1}{2} \|\boldsymbol{x} - \boldsymbol{u}\|^2 + \gamma h(\boldsymbol{x}) \right\}$$

where $h$ is a (possibly nonconvex) function defined in (7). Our objective is to demonstrate that $\boldsymbol{y}^* = \boldsymbol{u}$ is the unique stationary point and global minimizer of

$$\varphi(\boldsymbol{y}) := \frac{1}{2} \|\mathsf{D}_\sigma(\boldsymbol{y}) - \boldsymbol{u}\|^2 + \gamma h(\mathsf{D}_\sigma(\boldsymbol{y})), \quad \boldsymbol{y} \in \mathbb{R}^n.$$

By using the definition of $h$ in (7) and the Tweedie's formula (40), we obtain

$$\varphi(\boldsymbol{y}) = \frac{1}{2} \|\mathsf{D}_\sigma^*(\boldsymbol{y}) - \boldsymbol{u}\|^2 - \frac{\sigma^4}{2} \|\nabla h_\sigma(\boldsymbol{y})\|^2 + \sigma^2 h_\sigma(\boldsymbol{y}).$$

The gradient of $\varphi$ is then given by

$$\nabla \varphi(\boldsymbol{y}) = [\mathsf{JD}_\sigma(\boldsymbol{y})](\mathsf{D}_\sigma(\boldsymbol{y}) - \boldsymbol{u}) + \sigma^2 [\boldsymbol{I} - \sigma^2 \mathsf{H}h_\sigma(\boldsymbol{y})] \nabla h_\sigma(\boldsymbol{y}) = [\mathsf{JD}_\sigma(\boldsymbol{y})](\boldsymbol{y} - \boldsymbol{u}),$$

where we used (42) in the second line and (40) in the third line. Consider a scalar function $q(\nu) = \varphi(\boldsymbol{u} + \nu \boldsymbol{y})$ and its derivative

$$q'(\nu) = \nabla\varphi(\boldsymbol{u} + \nu\boldsymbol{y})^{\mathsf{T}}\boldsymbol{y} = \nu\boldsymbol{y}^{\mathsf{T}}[\mathsf{JD}_\sigma^*(\boldsymbol{u} + \nu\boldsymbol{y})]\boldsymbol{y}.$$

The positive definiteness of the Jacobian (42) implies that $q'(\nu) < 0$ and $q'(\nu) > 0$ for $\nu < 0$ and $\nu > 0$. Thus, $\nu = 0$ is the global minimizer of $q$. Since $\boldsymbol{y} \in \mathbb{R}^n$ is arbitrary, we can conclude that $\varphi$ has no stationary point other than $\boldsymbol{y}^* = \boldsymbol{u}$, and that $\varphi(\boldsymbol{u}) < \varphi(\boldsymbol{y})$ for any $\boldsymbol{y} \neq \boldsymbol{u}$ (Xu et al., 2020).

## F   ON THE ASSUMPTIONS OF THEOREM 1

In this section, we present the list of assumptions required for Theorems 1. Assumptions required for Theorems are typically employed when using MMSE estimators as PnP priors, engaging in nonconvex optimization, or dealing with mismatched/inexact PnP priors.

**Assumptions of Theorem 1:**

- *Prior distributions $p_{\boldsymbol{x}}$ and $\widehat{p}_{\boldsymbol{x}}$, denoted as target and mismatched distributions are non-degenerate over $\mathbb{R}^n$.*

  As discussed in Section 3.2, this assumption is commonly adopted to establish a relation between regularized inversion and MMSE estimation (Gribonval, 2011; Gribonval & Machart, 2013; Kazerouni et al., 2013). The MMSE estimators have been previously used as priors in PnP methods (Xu et al., 2020; Gan et al., 2023; Laumont et al., 2022).

- *Function $g$ (data-fidelity term) is continuously differentiable.*

  This assumption is an standard assumption commonly adopted in nonconvex optimization, specifically in the context of inverse problems (Li & Li, 2018; Jiang et al., 2019; Yashtini, 2021). It is worth noting that the majority of well-established data-fidelity terms for image restoration tasks fall under the umbrella of this assumption. Importantly, this framework does not necessitate the convexity of data-fidelity terms, making it versatile for handling non-linear measurement models. Furthermore, our result can be extended to a non-differentiable data-fidelity term $g$ by using subdifferentials, making it applicable to applications like phase retrieval (Metzler et al., 2018).

- *The explicit data-fidelity term $g$ and the implicit regularizer $h$ are bounded from below.*

  This assumption is standard in optimization and ensures that the optimization problem is well-posed and has a meaning full solution. This Assumption is commonly adopted in optimization algorithms (Yashtini, 2021; Hurault et al., 2022b;a; Xu et al., 2020).

- *The denoisers $\mathsf{D}_\sigma$ and $\widehat{\mathsf{D}}_\sigma$ have the same range $\mathsf{Im}(\mathsf{D}_\sigma)$. Additionally, functions $h$ and $\hat{h}$ associated with $\mathsf{D}_\sigma$ and $\widehat{\mathsf{D}}_\sigma$, are continuously differentiable with $L$-Lipschitz continuous gradients over $\mathsf{Im}(\mathsf{D}_\sigma)$.*

  For the image denoisers that share the same architecture and employ the same loss function, it is reasonable to assume that their output range would be consistent, given that it aligns with the range of natural color images. Furthermore, due to the smoothness properties of both $\mathsf{D}_\sigma^{-1}$ and $h_\sigma$ as described in equation 7, it follows that the function $h$ is also smooth and continuously differentiable. A similar property holds for the function $\hat{h}$ corresponding to the mismatched denoiser $\widehat{\mathsf{D}}_\sigma$. Consequently, this assumption is a mild requirement, only necessitating that regularization functions have $L$-Lipschitz continuous gradients over their shared range. While the assumption of Lipschitz continuous gradients is a standard one in nonconvex optimization, it is typically enforced over the entire space $\mathbb{R}^n$, whereas here, we specifically enforce it over the range of the denoisers. (Hurault et al., 2022a; Yashtini, 2021).

- *The distance between the target and mismatched denoisers are bounded at each iteration of the algorithm.*

  This assumption bounds the distance between the mismatched and target denoisers, which serves as a measure of the distribution shift. As the distributions used to train the mismatched denoisers diverge from the target distribution, we anticipate the bound on denoisers' distance will also increase. This assumption is a common one in the context of dealing with approximate, inexact, or mismatched priors (Laumont et al., 2022; Shoushtari et al., 2022; Gan et al., 2023).

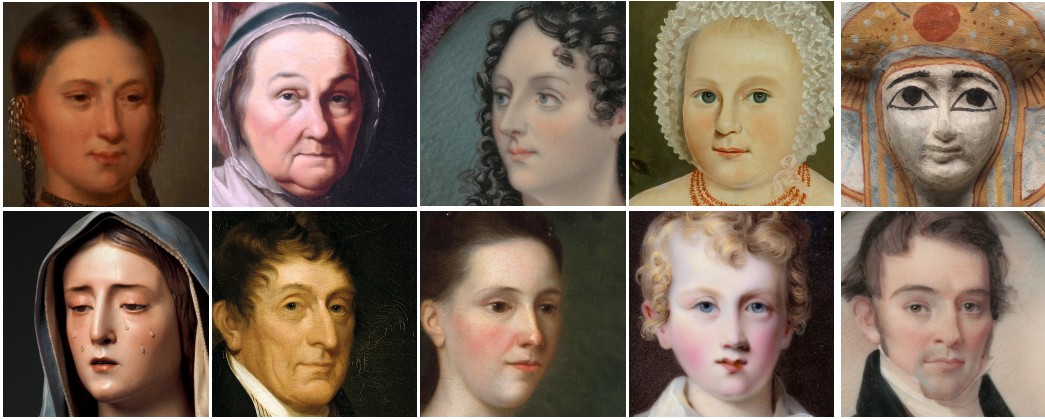

**Figure 6:** *Ground truth images from MetFaces dataset used for generating measurements.*

- *The distance of sequence $(\boldsymbol{z}^k)$ given by the Algorithm 1 to stationary point $\boldsymbol{z}^*$ is bounded by a constant.*

  As depicted in Algorithm 1, sequence $\boldsymbol{z}^k$ is the output of mismatched denoiser at each iteration. Since many denoisers have bounded range spaces, the existence of bound $R$ often holds. Specifically, this is true for such image denoisers whose output live within the bounded subset $[0, 255]^n \subset \mathbb{R}^n$ or $[0, 1]^n \subset \mathbb{R}^n$ (Sun et al., 2021; Sun et al., 2019).

## G  ADDITIONAL TECHNICAL DETAILS

We present here some technical details and results that were not included in the main paper. In our quantitative comparisons of different priors, we utilized the Peak Signal-to-Noise Ratio (PSNR) metric, which is defined as follows:

$$PSNR(\widehat{\boldsymbol{x}}, \boldsymbol{x}) = 20 \log_{10} \left( \frac{1}{\|\widehat{\boldsymbol{x}} - \boldsymbol{x}\|_2} \right),$$

where $\boldsymbol{x}$ represents the ground truth and $\widehat{\boldsymbol{x}}$ denotes the estimated image. Additionally, we include SSIM, a widely used metric in image processing and computer vision, to measure the similarity between two images. SSIM takes into account three components of an image: luminance, contrast, and structure. It compares local patterns of pixel intensities and is particularly useful for evaluating the perceived quality of compressed or processed images. For our PnP-ADMM algorithm, we performed 15 iterations for all denoisers. All denoisers (Adapted, matched, and mismatched) were trained using the DRUNet architecture (Zhang et al., 2021) with Mean Squared Error (MSE) loss, employing the Adam optimizer (Kingma & Ba, 2015) with a learning rate of $10^{-4}$. We incorporated a noise level map strength that decreases logarithmically from $\sigma_{\text{optim}}$ to $\sigma = 0.01$ over 15 iterations, where $\sigma_{\text{optim}}$ is fine-tuned for optimal performance for each test image and prior individually. To prepare the training and testing images from datasets such as MetFaces (Karras et al., 2020), AFHQ (Choi et al., 2020), CelebA (Liu et al., 2015), and RxRx1 (Sypetkowski et al., 2023), we randomly selected 1000 images and resized them to $256 \times 256$ slices. For the BreCaHAD (Aksac et al., 2019) dataset, we cropped the images to $512 \times 512$ and subsequently resized them to $256 \times 256$ slices for both the training and testing datasets.

Figure 6 shows the images that were used to generate measurements for super-resolution task.

## H  ADDITIONAL EXPERIMENTS

### H.1  SUPER-RESOLUTION

We present additional image super-resolution results for a more comprehensive understanding. Figure 7 illustrates the performance comparison of denoising and super-resolution using different

**Table 3:** Assumption Comparison in Convergence of PnP Methods

| Variant | data-fidelity | denoiser | mismatch (Y/N) |
|---|---|---|---|
| PnP-FBS (Ryu et al., 2019) | strongly convex | residual nonexpansive | ✗ |
| PnP-ADMM (Chan et al., 2017) | bounded gradient | bounded | ✗ |
| GS-PnP (Hurault et al., 2022a) | convex | gradient step | ✗ |
| PnP-PGM (Sun et al., 2019a) | convex | $\alpha-$averaged | ✗ |
| PnP-ADMM (Sun et al., 2021) | convex | residual nonexpansive | ✗ |
| RED (Shoushtari et al., 2022) | convex | nonexpansive | ✓ |
| PnP-ADMM (ours) | - | MMSE | ✓ |

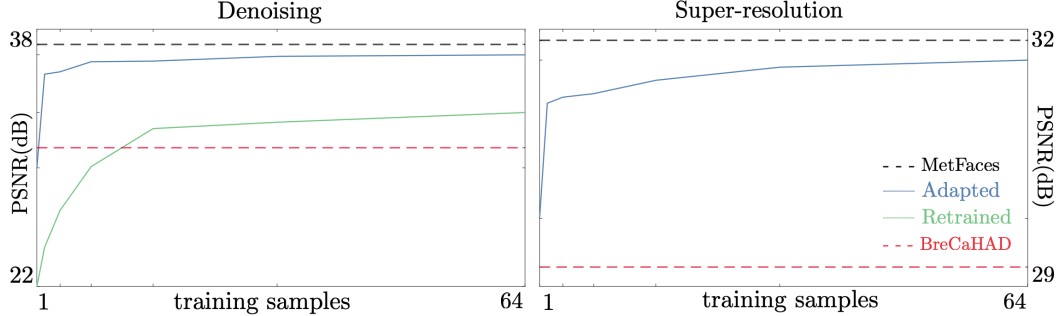

**Figure 7:** *The Left figure compares the empirical results of denoising for retrained and adapted priors vs. the number of training samples, as well as target (MetFaces) and mismatched (BreCaHAD) denoisers. The right figure compares PnP performance using target, mismatched, and adapted priors on super-resolution task. The results in both figures are reported for the test set from MetFaces dataset, averaged for scaling factor of $s = 4$. It's worth highlighting the noticeable performance improvement of denoisers achieved through domain adaptation. Additionally, observe the relationship between PnP performance and adapted denoiser performance.*

**Table 4:** *PSNR (dB) and SSIM values for image super-resolution using PnP-ADMM under different priors on a test set from the MetFaces (Karras et al., 2020) averaged for all kernels. We highlighted the **best** performing and the* worst *performing priors. BreCaHAD is the worst prior that is also the one visually most different from MetFaces. Measurement noise is set to 0.03.*

| Prior | $s = 2$ | | $s = 4$ | | Avg | |
|---|---|---|---|---|---|---|
| | PSNR | SSIM | PSNR | SSIM | PSNR | SSIM |
| BreCaHAD | 27.58 | 0.7214 | 24.79 | 0.6764 | 26.18 | 0.6989 |
| RxRx1 | 29.86 | 0.7599 | 28.14 | 0.7197 | 29.00 | 0.7398 |
| AFHQ | 30.04 | 0.7622 | 28.47 | 0.7194 | 29.34 | 0.7408 |
| CelebA | 30.11 | 0.7650 | 28.57 | 0.7235 | 29.34 | 0.7442 |
| MetFaces | **30.42** | **0.7754** | **28.88** | **0.7367** | **29.65** | **0.7560** |

priors. On the left side of Figure 7, the denoising performance of target (trained on MetFaces), mismatched (trained on BreCaHAD), adapted, and retrained priors is displayed. Meanwhile, on the right side, the reconstruction performance of target, mismatched, and adapted priors is presented. Note the improvement achieved by using adapted priors in both denoising and super-resolution tasks.

## H.2 DEBLURRING

We present additional visual results for deblurring image restoration. Figure 8 presents a visual comparison of a test image from the MetFaces dataset using the target denoiser and four different mismatched denoisers. The images are convolved with the indicated blur kernel and subjected to Gaussian noise with a noise level of $v = 0.01$. Note the suboptimal performance of mismatched priors in the deblurring task. As it is evident in Figure 8, the discrepancy between the mismatched distributions directly affects the PnP performance. Figure 9 illustrates a visual comparison for adapted priors in the deblurring task.

**Table 5:** *PSNR (dB) and SSIM comparison of super-resolution with mismatched, target, and adapted denoisers for the test set from MetFaces, averaged for all kernels. We highlighted the **target**, mismatched, and the best adapted priors. Measurement noise is set to 0.03.*

| Prior | $s = 2$ | | $s = 4$ | | $Avg$ | |
|---|---|---|---|---|---|---|
| | PSNR | SSIM | PSNR | SSIM | PSNR | SSIM |
| BreCaHAD | 27.58 | 0.7214 | 24.79 | 0.6764 | 26.18 | 0.6989 |
| 4 imgs | 30.03 | 0.7713 | 28.26 | 0.7319 | 29.14 | 0.7516 |
| 16 imgs | 30.36 | 0.7786 | 28.85 | 0.7411 | 29.60 | 0.7598 |
| 64 imgs | 30.39 | 0.7775 | 28.90 | 0.7410 | 29.64 | 0.7592 |
| MetFaces | 30.42 | 0.7754 | 28.88 | 0.7367 | 29.65 | 0.7560 |

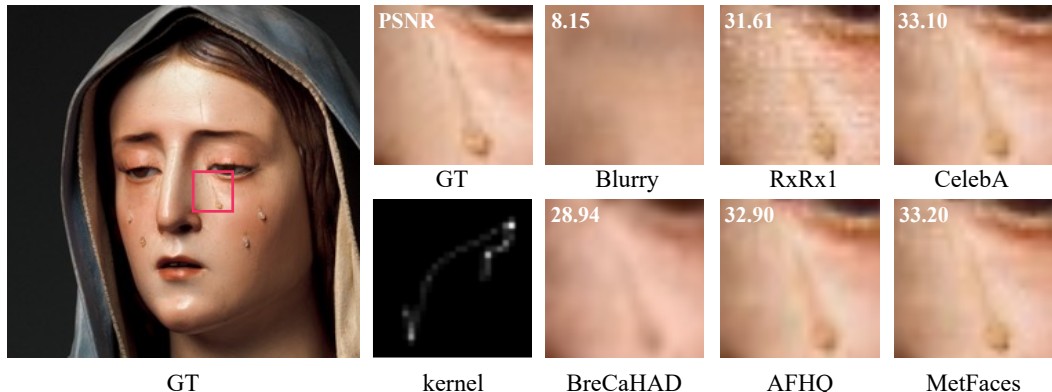

**Figure 8:** *Visual comparison of various mismatched denoisers for deblurring on an image from MetFaces dataset. The performance is reported in terms of PSNR (dB). The image is convolved with the indicated blur kernel and Gaussian noise with $v = 0.01$ is added. Note that regardless of the PnP image restoration task, the discrepancies in training distributions result in mismatched priors and suboptimal performance in PnP.*

### H.3 PHASE RETRIEVAL

We evaluate the performance of PnP-ADMM in addressing a nonconvex phase retrieval problem involving *coded diffraction patterns (CDP)*, a scenario akin to those explored in prior studies (Metzler et al., 2018; Wu et al., 2019). The object $x \in \mathbb{R}^n$ is exposed to illumination from a coherent light source. A random known phase mask, represented by $M$, modulates the light, with each entry of $M$ randomly drawn from the unit circle in the complex plane. The light undergoes far-field Fraunhofer diffraction, and a camera captures its intensity as $y \in \mathbb{R}_+^m$. Given the Fraunhofer diffraction's representation through a Fourier Transform, the data-fidelity term for this phase reconstruction problem is expressed as follows:

$$g(x) = \frac{1}{2}\|y - |FMx|\|_2^2, \tag{43}$$

where $F$ denotes the 2D discrete Fast Fourier Transform (FFT). Figure 10 illustrates the performance of mismatched, target and adapted priors in the problem phase retrieval using PnP-ADMM. Note that adapting to larger set of paired data from target domain can effectively close the performance gap. Table 6 reports numerical results achieved using PnP-ADMM with matched, mismatched, and target priors for different input SNR, averaged for MetFaces testset.

### H.4 VARIOUS DISTRIBUTIONS EXPERIMENT

We present additional visual results for mismatched priors and domain adaptation using various distributions for image super-resolution. In the following Figures, we demonstrate the effect of mismatched priors and prior adaptation tested on an image from RxRx1 (Sypetkowski et al., 2023) dataset. Figure 11 presents a visual comparison for PnP on super-resolution task using the target and three mismatched priors on an image from the RxRx1 test set. The images are convolved with the blur

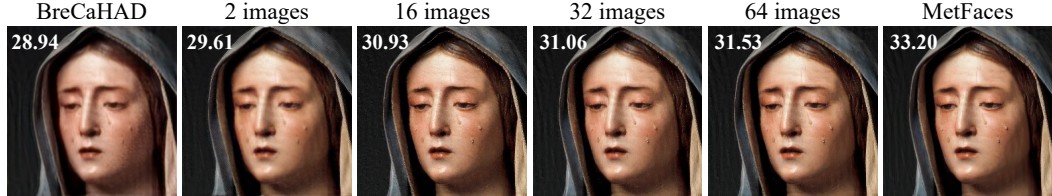

**Figure 9:** *Visual comparison of several adapted prior for image deblurring on a test image from MetFaces dataset. The performance is reported in terms of PSNR (dB). The experiment setting is similar to that of Figure 8. Note how adapting the mismatched prior with a larger set of data from the target distribution results in a better performance in PnP.*

**Table 6:** *PSNR (dB) and SSIM comparison of phase retrieval problem with mismatched, target, and adapted denoisers for the test set from MetFaces, for various input SNR. We highlighted the **target**, mismatched, and the best adapted priors.*

| Prior | InputSNR= 15 | | InputSNR= 20 | | InputSNR= 25 | |
|---|---|---|---|---|---|---|
| | PSNR | SSIM | PSNR | SSIM | PSNR | SSIM |
| BreCaHAD | 24.13 | 0.7491 | 25.86 | 0.8278 | 26.53 | 0.8584 |
| AFHQ | 27.35 | 0.8086 | 29.75 | 0.8760 | 30.76 | 0.8967 |
| MetFaces | **27.57** | **0.8123** | **29.88** | **0.8798** | **31.13** | **0.9005** |
| 4 imgs | 26.47 | 0.7833 | 28.80 | 0.8616 | 29.95 | 0.8904 |
| 16 imgs | 26.84 | 0.7930 | 29.21 | 0.8673 | 30.34 | 0.8926 |
| 64 imgs | 27.18 | 0.8015 | 29.60 | 0.8731 | 30.90 | 0.8983 |

kernel indicted in Figure 3. Figure 12 illustrates visual results for domain adaptation of mismatched prior trained on CelebA dataset and adapted to RxRx1 distribution. Note the improvement in PnP performance by using adapted priors. Also, note the relation between PnP performance and the number of samples from the target distribution used for adaptation.

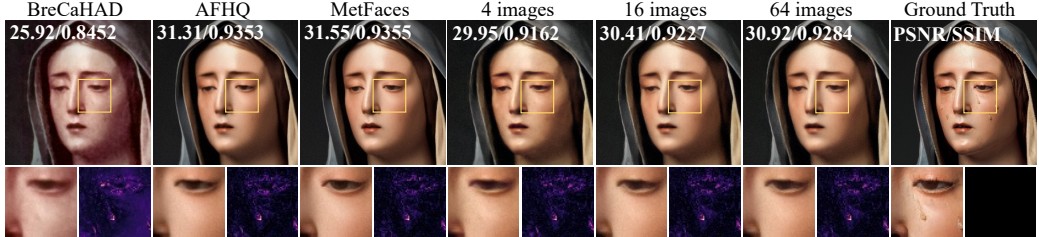

**Figure 10:** *Visual comparison of several adapted and mismatched priors for phase retrieval problem on a test image from MetFaces dataset. The performance is reported in terms of PSNR (dB) and SSIM. Note how adapting the mismatched prior with a larger set of data from the target distribution results in a better performance in PnP.*

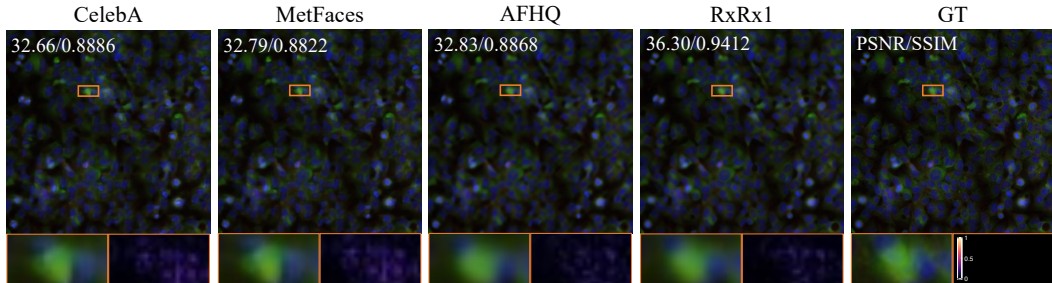

**Figure 11:** *Visual evaluation of several priors on the image super-resolution task reported in terms of PSNR (dB) and SSIM for an image from RxRx1. Images are downsampled with the scale of $s = 4$ and convolved with the indicated blur kernel in Figure 3. Note the influence of mismatched priors on the performance of PnP.*

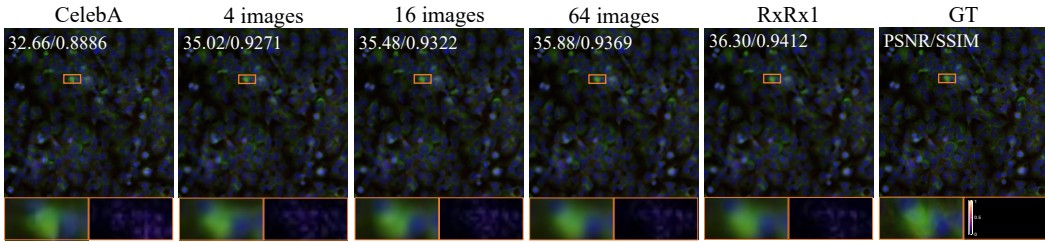

**Figure 12:** *Visual comparison of image super-resolution with target (RxRx1), mismatched (CelebA), and adapted priors on a test image from RxRx1. The images are downsampled by the scale of $s = 4$. The performance is reported in terms of PSNR (dB) and SSIM. Note how the recovery performance increases by adaptation of mismatched priors to a larger set of images from the target distribution.*

