# OpenReview forum: "Prior Mismatch and Adaptation in PnP-ADMM with a Nonconvex Convergence Analysis"
_ICLR.cc/2024/Conference — Submitted to ICLR 2024_

### Official Review · Reviewer_aNsR · 2023-10-28

**Soundness:** 3 good
**Presentation:** 3 good
**Contribution:** 3 good
**Rating:** 8
**Confidence:** 4

**Summary:**

This paper studies the behavior of plug-and-play priors with the alternating directions method of multipliers (ADMM) in case there is a mismatch in the distribution of the data the denoising prior has been trained on and the data it is applied to. The authors assume the denoisers to be perfect  minimum mean squared error (MMSE) estimators and prove convergence of the plug-and-play prior ADMM under this (as well as a few more technical) assumption. Moreover, they provide an estimate for how much the gradients of the underlying objective  function corresponding to the ideal MMSE prior differ from zero if a prior trained on a different distribution is used. Numerical experiments demonstrate (not surprisingly) that using prior trained on a different distribution can lead to suboptimal results and that (surprisingly) as few as 4 images are enough to fine-tune the denoiser to yield significantly better results.

**Strengths:**

- The paper conducts an interesting theoretical analysis on a problem that has received comparably little attention in the literature so far.
- It demonstrates numerically that even a small number of images can help to fine-tune (domain-adapt) a denoiser such that it performs significantly better within the plug-and-play framework in case of domain shifts.

**Weaknesses:**

- At first, I just wanted to state that the theorems are impossible: As $(x^0, z^0, s^0)$ is the (arbitrary) starting point of the ADMM algorithm, it is impossible to ensure that $\phi(x^0, z^0, s^0) - \phi^* \geq 0$, which immediately contradicts the inequality in Theorem 2. Yet, I think this is just a typo, i.e., one needs to start with the first iteration of the ADMM algorithm to make the theorem work. I'd like to ask the authors to briefly confirm this as I have not checked any proof.
- I cannot judge the contribution on the theoretical side very well yet. I phrased a question on this below.
- The practical implications of the paper are not large (= it is good to know that a setting similar to what everyone does in practice converges, but the assumptions are of course never verifiable exactly; knowing a constant by which the gradient of the objective differs from zero still makes it hard to say something about the point one converges to).

Yet, the third point is not uncommon for theory-inspired papers and would not keep me from recommending the acceptance of this work.

**Questions:**

- Even after realizing that the theorems' starting points might just have been a typo, the results still appeared to be a little surprising (as the augmented Lagrangian is not monotone in the usual convex ADMM setting). Then, I, however, remembered the paper "Global Convergence of ADMM in Nonconvex Nonsmooth Optimization", where the augmented Lagrangian was also used as a Lyapunov function (which might therefore be worth citing). How similar/different is the conducted convergence analysis from taking existing literature on the analysis of nonconvex ADMM (the one I cited, versions that make smoothness assumptions, and those that have possibly considered an inaccurate evaluation of the prox) and just making assumptions that allow you to return to such a known analysis?  Things like 'summable errors' in substeps of such algorithms are also very common in the general optimization literature.
- The paragraph below Theorem 2 speaks of convergence to a stationary point. Convergence is, however, not what the theorems state. Is this something that comes out of the actual proof in the supplement? If so, it would be worth stating in the main paper. If not, please rephrase.
- In assumption 6, one needs the inequality to be satisfied for all stationary points of the augmented Lagrangian, and the theorems are to be interpreted as "there exists $(x^*, z^*, s^*)$ such that ...", correct? Does the existence of a stationary point have to be assumed or do the assumptions guarantee that?
- Can you please comment on assumption 1: Isn't it common to assume that "realistic" images lie in some set (e.g. a union of subspaces) much smaller than the dimension of the space? Would that cause a problem for assumption 1?

---

> ### Author Response · Authors · 2023-11-21
> **Response to Reviewer aNsR**
>
> Thank you for your feedback and positive comments on our work. Please see below for our point-by-point responses to your comments.
>
> >At first, I just wanted to state that the theorems are impossible: As $ϕ(x^0,z^0,s^0)$ is the (arbitrary) starting point of the ADMM algorithm, it is impossible to ensure that $ϕ(x^0,z^0,s^0)−ϕ^∗\geq0$, which immediately contradicts the inequality in Theorem 2...
>
> Thank you for pointing out this inconsistency in our notations. We have revisited the statements for our assumptions, theorems, and proofs to streamline the presentation. The updated theorems statement combines all the constants into $A_1$ and $A_2$, while the proof explicitly shows that $ϕ(x^0,z^0,s^0)−ϕ^∗\geq0$, by using Lemma 3 that shows that the augmented Lagrangian monotonically decreases for the iterations of PnP-ADMM.
>
> >The practical implications of the paper are not large...
>
> Indeed, we revised the discussion after our theorems to make the statements more precise.
>
> >Even after realizing that the theorems' starting points might just have been a typo, the results still appeared to be a little surprising (as the augmented Lagrangian is not monotone in the usual convex ADMM setting). Then, I, however, remembered the paper "Global Convergence of ADMM in Nonconvex Nonsmooth Optimization"...
>
> Thank you for providing [1], which we have cited in the revised manuscript. The significance of our analysis lies in integrating the concepts and assumptions from PnP to obtain a novel analysis of corresponding ADMM iterations. The traditional literature on ADMM does not leverage the specific properties of PnP operators, while the current PnP literature has not considered a nonconvex analysis of PnP-ADMM under mismatched deep denoisers. This allows us to bring unique insights on a popular topic within the imaging inverse problems community.
>
> [1] Wang, Yu, Wotao Yin, and Jinshan Zeng. "Global convergence of ADMM in nonconvex nonsmooth optimization." Journal of Scientific Computing 78 (2019): 29-63.
> >The paragraph below Theorem 2 speaks of convergence to a stationary point. Convergence is, however, not what the theorems state. Is this something that comes out of the actual proof in the supplement? If so, it would be worth stating in the main paper. If not, please rephrase.
>
> The revised paragraph clarifies this better. The convergence of the gradient to zero is only true when errors are summable. When not, the convergence is not guaranteed, but one can still obtain a bound on the norm of the gradient given in the theorems. The proof naturally provides more details by showing the residual convergence for summable errors. Since these details won’t provide new insights into the mismatch, we leave them in the supplement.
>
> >In assumption 6, one needs the inequality to be satisfied for all stationary points of the augmented Lagrangian, and the theorems are to be interpreted as "there exists $(x^∗,z^∗,s^∗)$ such that ...", correct? Does the existence of a stationary point have to be assumed or do the assumptions guarantee that?
>
> We appreciate the reviewer's feedback. We have updated the manuscript to streamline the presentation of our results and show the existence of the infimum of the augmented Lagrangian.
>
> >Can you please comment on assumption 1: Isn't it common to assume that "realistic" images lie in some set (e.g. a union of subspaces) much smaller than the dimension of the space? Would that cause a problem for assumption 1?
>
> While the “union of subspaces” model is not compatible with our Assumption 1, there are many other “sparse” models that are. Any distribution that has density everywhere, even when that density has very low probability works. For example, Assumption 1 is compatible with the assumption that image coefficients are distributed according to the Bernoulli-Gaussian distribution (i.e., either zero or Gaussian coefficients). It is also worth noting that Assumption 1 can be relaxed to generalize the theory to arbitrary priors (see Remark 2 in (Gribonval, 2011)). The benefit of the current presentation is the simplicity, which doesn’t really reduce generality.

---

> > ### Comment · Reviewer_aNsR · 2023-11-22
> > **Thanks**
> >
> > Thank you for the answers. Please still go over the supplementary material one more time. There still appears $(x^0, z^0, s^0)$ (e.g. in eq. (30)), which -- looking at your algorithm 1 -- is not defined (as there is no $x^0$, or alternatively, $x^0$ can be anything such that it is hard to imagine any inequality can be shown). Please make sure your superscripts only occur for $k\geq 1$ (at least for $x^k$). Also please check the definition of $B$ below eq. (30).
> >
> > Finally, if the idea of the convergence proof is to use the augmented Lagrangian as a Lyapunov functional, I'd highly recommend stating that. It is little extra text but could help the reader significantly.

---

> > > ### Author Response · Authors · 2023-11-22
> > > **Response to Reviewer aNsR**
> > >
> > > >Please still go over the supplementary material one more time. There still appears $(x^0,z^0,s^0)$ (e.g. in eq. (30)), which -- looking at your algorithm 1 -- is not defined...
> > >
> > > We express our gratitude for the valuable feedback provided by the reviewer. Assumption 3, which addresses the boundedness of $f$ for all $x \in\mathbb{R}^n$, including $x^0$, is used to establish the boundedness of the augmented Lagrangian. We have included a discussion specific to $x^0$ within  Lemma 2 for additional clarification.
> > >
> > > >Also please check the definition of $B$ below eq. (30).
> > >
> > > We appreciate the clarification regarding the definition of B, which we have updated accordingly.
> > >
> > > >Finally, if the idea of the convergence proof is to use the augmented Lagrangian as a Lyapunov functional, I'd highly recommend stating that. It is little extra text but could help the reader significantly.
> > >
> > > We also have included this in the paragraph after the theorems.

---

### Official Review · Reviewer_rsM4 · 2023-10-30

**Soundness:** 3 good
**Presentation:** 4 excellent
**Contribution:** 2 fair
**Rating:** 6
**Confidence:** 4

**Summary:**

This paper investigates the impact of mismatched denoisers in PnP-ADMM, offering theoretical insights and empirical validation on image super-resolution while introducing a domain adaptation strategy to mitigate distribution mismatch effects, i.e. distribution alignment. The convergence analysis of this work does not require convex data-fidelity and non-expansive denoiser assumptions.


------------------------------------------post-rebuttal------------------------------------------

Thank you for your thorough responses. I've reviewed your replies to my questions and the revised paper. The changes made in the revised paper have contributed to a better understanding of your work and I'm satisfied with the changes made. I am inclined to reassess the paper positively and recommend acceptance.

**Strengths:**

Novelty: A theoretical error bound for PnP-ADMM under mismatched denoisers is derived, without the constraints of convex data-fidelity and non-expansive denoiser assumptions.

Significance: It shows the pivotal role of domain adaptation in mitigating the effects of distribution shifts on image priors.

**Weaknesses:**

Experimental Limitations: The paper's experiments primarily focus on super-resolution and deblurring with negligible measurement noise, limiting the comprehensiveness of the investigations. It lacks exploration of scenarios with nonconvex data-fidelity terms, which would enrich the analysis and diversify the paper's empirical scope.

Inadequate Literature Review: The related work presented in the paper lacks depth and breadth (e.g., in domain adaptation, and review of previous work).

**Questions:**

The paper in question would benefit from addressing several issues:

    1. The paper incorrectly describes PnP as a class of deep learning algorithms (e.g., “PnP has emerged as a class of DL algorithms”, “MBDL approaches include methods such as PnP, RED, …”, “PnP is one of the most popular MBDL approaches for solving image inverse problems”). It is essential to clarify that PnP is an iterative framework in which a denoiser can be incorporated as an implicit prior model. The presence of a deep denoiser within PnP does not classify it as a class of deep learning algorithms.

    2. In the second paragraph of Page 1, it is advisable to provide a brief introduction to PnP, with more detailed information presented in the Background section.

    3. On Page 1, the sentence “Despite extensive literature on PnP, the research in the area has mainly focused on the setting where the distribution of the test or inference data is perfectly matched to that of the data used for training the image denoiser” should be rewritten to specifically refer to approaches utilizing deep learning-based denoisers within PnP.

    4. The sentence on Page 1, “Little work exists for PnP under mismatched priors” should be substantiated with appropriate references, such as (Shoushtari et al., 2022), and other relevant works. It is recommended to create a subsection or, at the very least, a paragraph discussing the similarities and differences between the present work and that of (Shoushtari et al., 2022), aside from variations in the PnP framework and applications.

    5. On Page 1, in the sentence “The success of PnP has resulted in denoising diffusion probabilistic models”, it is advisable to cite each topic individually, as appropriate.

    6. It is suggested to briefly define “domain adaptation” in the Introduction, considering that the term is used five times before its formal definition in Section 4.2. Additionally, different strategies in domain adaptation should be mentioned (e.g., see 10.1109/CVPR.2014.187 and 10.1109/ACCESS.2019.2929258) to provide context for the selected ‘distribution alignment’ strategy and to explain why it was selected. Please also explain how a limited number of data from target distribution can address the mismatched prior distribution.

    7. On Page 2, considering the provided definition of model-based deep learning (DL) as “methods that integrate the measurement model as part of the deep model”, it is necessary to clarify how employing DL-based denoisers within PnP/RED aligns with this definition. The relationship, if any, between model-based DL, model-based iterative reconstruction/recovery, and physics-informed DL should also be addressed.

    8. On Page 2, the references cited after “DL methods seek to perform a regularized inversion … by a deep convolutional neural network (CNN)” are irrelevant to the context of this sentence and can be omitted. Instead, references such as (McCann et al., 2017; Lucas et al., 2018; Ongie et al., 2020; Monga et al., 2021) are more suitable in this context.

    9. A table of assumptions for the cited references regarding the theoretical convergence analyses of PnP should be considered. Such a table would be valuable for readers to understand the differences between these works and the current study.

    10.  The experiments detailed in Section 4 raise a query: is the measurement noise level consistently set at 0.01? If so, this level of noise appears comparable to the noise-free scenario. It would significantly enhance the paper to explore the impact of more robust and varying levels of measurement noise on the obtained results, underlining the model's performance in noisier conditions.

    11.  While the experiments concentrate on convex data fidelity terms using the $\ell_2$ norm, it would be advantageous to broaden the scope by including experiments with nonconvex data-fidelities. Extending the experiments to encompass nonconvex scenarios would further corroborate the claims made in the paper.

    12.  The paper presents Theorems 1 and 2, providing theoretical support for the proposed methodologies. It would be beneficial to provide a descriptive narrative of the results or outcomes derived from these theorems to facilitate a more comprehensive understanding for readers.

    13.  Sections A-D of the Appendices delve into the topic of “the optimality condition for mismatched MMSE denoiser.” Further explanation is required to enhance the clarity and comprehension of this topic.

    14.  A point of inquiry arises in relation to Page 15 (above Eq.(13)) regarding the derivation of $\nabla\hat{h}(z^k)=\frac{1}{\gamma}(s^{k-1}+x^k-z^k)=\frac{1}{\gamma}s^k$. Similarly, clarity on the derivation of Eq.(16) on Page 16 would fortify the readability and comprehensibility of the paper.

    15.  An observation on the gradient of h in Eq.(16) leads to a query: shouldn't the gradient of Eq.(18) be zero? As $\nabla \frac{1}{2\gamma}\|z-(x^k+s^{k-1})\|_2^2 = \frac{1}{\gamma}(z-(x^k+s^{k-1}))$ and $\nabla h(z) = \frac{1}{\gamma} (x^k+s^{k-1}-z)$.

---

> ### Author Response · Authors · 2023-11-21
> **Response to Reviewer rsM4**
>
> Thank you for your feedback and positive comments on our work. Please see below for our point-by-point responses to your comments.
>
> >Experimental Limitations: The paper's experiments primarily focus on super-resolution and deblurring with negligible measurement noise, limiting the comprehensiveness of the investigations. It lacks exploration of scenarios with nonconvex data-fidelity terms, which would enrich the analysis and diversify the paper's empirical scope.
>
> Prompted by your comment, we have included the results for the phase retrieval problem, which is nonconvex, in Section H.3 of the supplementary materials.
>
> >Inadequate Literature Review: The related work presented in the paper lacks depth and breadth (e.g., in domain adaptation, and review of previous work).
>
> We have added a paragraph to the Background Section in Page 3 of the main paper, addressing domain adaptation in imaging inverse problems.
>
> **Questions**
> 1. Prompted by your comment, we revised the manuscript to reflect this point.
> 2. We revised the manuscript accordingly.
> 3. We revised the manuscript accordingly.
> 4. We have included additional discussion to paragraph 4 in Section 2 regarding the existing work in the revised manuscript.
> 5. Edited as suggested.
> 6. Thank you for providing the references; we have included them in paragraph five of Section 2, dedicated to domain adaptation.
> 7. Edited as suggested (see “Plug-and-Play Priors” in “Background”)
> 8. Edited as suggested.
> 9. Prompted by your comment, we have added Table 3 to Section G of the Supplementary materials for a comparison of existing theoretical analyses of PnP methods.
> 10. Prompted by the feedback, we have added Tables 4 and 5 to Section H.1 of the Supplementary material, presenting super-resolution results obtained with a measurement noise set to 0.03.
> 11. Prompted by feedback, we have included the results for a new phase retrieval experiment with a nonconvex data-fidelity term, in Section H.3 of the Supplementary materials.
> 12. We have adjusted the last two paragraphs of Section 3.1 to better clarify our results.
> 13. We have added additional clarification to the revised manuscript.
> 14. The revised manuscript includes an additional explanation of how to derive Equation 16.
> 15. The gradient of Equation (18) is zero only for $ \overline{z}^k$, a fact utilized in the subsequent equations on the same page, specifically the one preceding Equation (19).

---

> ### Author Response · Authors · 2023-11-23
> **Response to Reviewer rsM4**
>
> Dear Reviewer, thank you for reading our paper. We are nearing the end of the discussion period. Let us know if you have any additional questions that could further improve your evaluation of our paper.

---

### Official Review · Reviewer_E3jh · 2023-10-31

**Soundness:** 3 good
**Presentation:** 4 excellent
**Contribution:** 2 fair
**Rating:** 5
**Confidence:** 2

**Summary:**

The authors proposed theoretical analysis of discrepancy between the desired and mismatched denoisers in PnP-ADMM and can also applicable in non-convex data-fidelity terms and non-expansive denoisers. The experiment show that using a simple domain adaptation method can addressing the problem in PnP-ADMM on several super-resolution and deblurring datasets.

--- Post-rebuttal
I read the rebuttal and would like to keep the score.

**Strengths:**

The authors theoretically proves the gap between the denoiser in PnP-ADMM, including limitation and application boundaries.

**Weaknesses:**

1. The gap of pnp denoiser is a common view for many years, a series of work aiming to solve this problem are proposed like [1,2], but the authors do not compare even a single method.
[1] Shocher, Assaf, Nadav Cohen, and Michal Irani. "“zero-shot” super-resolution using deep internal learning." Proceedings of the IEEE conference on computer vision and pattern recognition. 2018.
[2] Tirer, Tom, and Raja Giryes. "Super-resolution via image-adapted denoising CNNs: Incorporating external and internal learning." IEEE Signal Processing Letters 26.7 (2019): 1080-1084.

2. The details of domain adaptation method are not described such as loss function and training time. The computational resources of domain adaptation process are not mentioned either. Usually, the training need 2-3 times larger GPU memory than simply implement inference. Additional data and computational resources for training may not appropriate for the pnp method which is designed for non supervision.

**Questions:**

1. Only SR and deblurring are selected as example tasks. Some more complex inverse problem such as the compressive sensing and phase retrieval?

2. What if the there is no ground truth in a real-world test dataset? Does it mean that we have to manually label find several images which are close to the test set?

---

> ### Author Response · Authors · 2023-11-21
> **Response to Reviewer E3jh**
>
> Thank you for your feedback and positive comments on our work. Please see below for our point-by-point responses to your comments.
>
> **Domain adaptation in PnP.** Prompted by your remark, we have added paragraph five in Section 2 to provide background on domain adaptation in imaging inverse problems, citing both [1] and [2]. As noted by the Reviewer, prior work has indeed considered model adaptation in the context of inverse problems; however, our work is novel since it is the first to explicitly relate a theoretical error bound within PnP-ADMM in a nonconvex setting to domain adaptation. None of the papers mentioned by the reviewer have considered this setting.
>
> [1] Shocher, Assaf, Nadav Cohen, and Michal Irani. "“zero-shot” super-resolution using deep internal learning." Proceedings of the IEEE conference on computer vision and pattern recognition. 2018.
>
> [2] Tirer, Tom, and Raja Giryes. "Super-resolution via image-adapted denoising CNNs: Incorporating external and internal learning." IEEE Signal Processing Letters 26.7 (2019): 1080-1084.
>
> **Model adaptation details.** Section G of the Supplementary material provides details on training and adaptation. The following table presents the time required for both training and adaptation.
>
>
> **Approximate training time (hours)**
> | ||||||
> |:-----------:|:-------------:|:------------:|:-------------:|:-------------:|:---------------:|
> |MetFaces &nbsp;|BreCaHAD &nbsp; |Adapted 4 &nbsp;|Adapted 16 &nbsp;|Adapted 32 &nbsp;| Adapted 64 &nbsp;|
> |14|15|0.05|0.10|0.20|0.33|
> | ||||||
>
> Note how pre-training image denoisers require a long training time; on the other hand, model adaptation requires much less time. The computational cost justifies using model adaptation of mismatched denoisers instead of training them from scratch.
>
> **Nonlinear inverse problems.**  Our theoretical assumptions are suitable for inverse problems with nonconvex data-fidelity terms. Prompted by your comment, Section H.3 in the Supplementary materials presents results on phase retrieval, using both adapted and pre-trained models for the MetFaces dataset.
>
> **No paired data.** Both theoretical and experimental results suggest that for closer data distributions, we get lower reconstruction errors. One could thus use a dataset closest to the inference dataset to fine-tune the pre-trained denoisers to achieve lower reconstruction error.

---

> ### Author Response · Authors · 2023-11-23
> **Response to Reviewer E3jh**
>
> Dear Reviewer, thank you for reading our paper. We are nearing the end of the discussion period. Let us know if you have any additional questions that could further improve your evaluation of our paper.

---

### Official Review · Reviewer_wzA2 · 2023-11-06

**Soundness:** 3 good
**Presentation:** 2 fair
**Contribution:** 3 good
**Rating:** 6
**Confidence:** 2

**Summary:**

This article proposes a theoretical analysis of PnP-ADMM algorithm under a mismatch on the prior.

**Strengths:**

The study of prior mismatch in inverse problems is a timely topic. This paper provides an explicit error bound on the solution of PnP-ADMM, which is challenging, and can be inspiring for the inverse problem community. The numerical experiments are well done can suggests an improvement with respect to the current literature.

**Weaknesses:**

The reviewer regrets the lack of contextual explanations on the result presented in Theorem 1, as well as of discussion on the significance of the performance metrics used in the analysis. The numerical simulation do not highlight the performance of the reconstruction of the proposed method with respect to the error bound proposed in Theorem 1.

**Questions:**

N/A

---

> ### Author Response · Authors · 2023-11-21
> **Response to Reviewer wzA2**
>
> >The reviewer regrets the lack of contextual explanations on the result presented in Theorem 1, as well as of discussion on the significance of the performance metrics used in the analysis. The numerical simulation do not highlight the performance of the reconstruction of the proposed method with respect to the error bound proposed in Theorem 1.
>
> Thank you for your feedback and positive comments on our work. Please see below for our point-by-point responses to your comments.
>
> **Discussion of Theorem 1.** We have extended the discussion in the last two paragraphs of Section 3.1 to provide more contextual explanation of the theoretical results. Theorem 1 is directly related to our experiments, since the reduction of $\delta_k$ via domain adaptation, directly ensures the decrease of the error term $\epsilon$ in PnP-ADMM.
>
> **Performance metrics.** Additional clarification regarding the quality metrics has been added to Section G of the Supplementary material.

---

> ### Author Response · Authors · 2023-11-23
> **Reviewer wzA2**
>
> Dear Reviewer, thank you for reading our paper. We are nearing the end of the discussion period. Let us know if you have any additional questions that could further improve your evaluation of our paper.

---

### Author Response · Authors · 2023-11-21
**Response to All Reviewers**

Thank you all for providing us with valuable feedback. We provide detailed answers to all the comments below. To better address some of them, we ran additional experiments. Most of these results were included in the dedicated sections of the revised supplementary material. The revised part in the main paper is indicated with blue highlighting.

---

### Author Response · Authors · 2023-11-23
**Response to all reviewers and area chairs**

Dear all, thank you for reading our paper and providing feedback. We are nearing the end of the discussion period. We have responded to all the comments. We hope that our responses will help other reviewers to also see the value in our work.

---

### Meta-Review · Area_Chair_5hpE · 2023-12-07

**Metareview:**

This paper studies PnP-ADMM when there is a mismatched denoiser. The main contribution of the paper is the convergence analysis of the PnP-ADMM under this setting, other contributions are numerical confirmation/illustration of this. The analysis is in fact about an inexact nonconvex ADMM, where the denoising step has an error. The authors require Assumption 5 to control this error, and the error is further required to be summable in order to guarantee the convergence. However, this assumption is very strong and is not checkable in practice. The discussions are not convincing why this is a reasonable assumption in practice. Moreover, after making this assumption, the convergence analysis becomes a simple application of existing theory on inexact nonconvex ADMM. In fact, one only needs to apply the convergence analysis of ADMM and whenever the mismatch comes to play, it is replaced by the target denoiser and the error. The analysis of ADMM for target denoiser is then applied (which has been established in the literature). Since this theoretical analysis is the main contribution of this paper, it is not significant enough to warrant acceptance.

**Justification For Why Not Higher Score:**

Results are incremental.

**Justification For Why Not Lower Score:**

NA

---

### Decision · Program_Chairs · 2024-01-16

Reject